# Boosting Skeleton-based Zero-Shot Action Recognition with Training-Free Test-Time Adaptation

**Jingmin Zhu**
Monash University
jingmin.zhu1@monash.edu

**Anqi Zhu**
Monash University
maggie.zhu@monash.edu

**Hossein Rahmani**
Lancaster University
h.rahmani@lancaster.ac.uk

**Jun Liu**
Lancaster University
j.liu81@lancaster.ac.uk

**Mohammed Bennamoun**
University of Western Australia
mohammed.bennamoun@uwa.edu.au

**Qiuhong Ke**[*]
Monash University
Qiuhong.Ke@monash.edu

## Abstract

We introduce *Skeleton-Cache*, the first training-free test-time adaptation framework for skeleton-based zero-shot action recognition (SZAR), aimed at improving model generalization to unseen actions during inference. Skeleton-Cache reformulates inference as a lightweight retrieval process over a non-parametric cache that stores structured skeleton representations, combining both global and fine-grained local descriptors. To guide the fusion of descriptor-wise predictions, we leverage the semantic reasoning capabilities of large language models (LLMs) to assign class-specific importance weights. By integrating these structured descriptors with LLM-guided semantic priors, Skeleton-Cache dynamically adapts to unseen actions without any additional training or access to training data. Extensive experiments on NTU RGB+D 60/120 and PKU-MMD II demonstrate that Skeleton-Cache consistently boosts the performance of various SZAR backbones under both zero-shot and generalized zero-shot settings. The code is publicly available at https://github.com/Alchemist0754/Skeleton-Cache.

## 1 Introduction

Action recognition is an important research area in computer vision with diverse applications in surveillance, monitoring, and human-computer interaction [12, 13]. While traditionally relying on RGB-based approaches, the field has evolved to include skeleton-based methods, enabled by advances in pose estimation techniques and depth sensor technologies [21, 27]. Skeleton-based recognition offers distinct advantages: robustness to lighting and background variations, enhanced privacy protection, and reduced computational demands [2, 14, 29, 23, 22]. These benefits have established it as a viable alternative to traditional video-based recognition, achieving competitive accuracy while addressing common challenges [25]. However, existing supervised skeleton-based methods rely on extensive labeled training data, which is often expensive and impractical to obtain, particularly for rare actions [5].

To address this limitation, Skeleton-based Zero-Shot Action Recognition (SZAR) [17, 1] has emerged, enabling recognition of unseen actions through supporting information such as class names and

---

[*]Corresponding author.

39th Conference on Neural Information Processing Systems (NeurIPS 2025).

attributes. The core idea aligns skeleton features with textual descriptors in a shared semantic space, transferring knowledge from known to unknown classes [9, 28, 32, 33].

In SZAR, models often face a significant distribution shift between training (seen actions) and testing (unseen actions) due to semantic gap between action categories, variations in human pose dynamics, differences in execution styles across individuals, and domain-specific noise such as sensor errors or occlusions. Moreover, since unseen actions may involve novel motion patterns or compositional pose structures not observed during training, the model's learned representations can fail to generalize. Test-time adaptation (TTA) is a compelling solution to this challenge—it allows models to adjust to the test data distribution on the fly, improving generalization without requiring retraining from scratch. However, conventional TTA [26, 24] methods often rely on gradient-based updates or additional optimization, which can be computationally expensive and unstable. While recent training-free alternatives [10, 31] have emerged, they are primarily designed for image classification tasks and are not readily applicable to SZAR, given the unique spatio-temporal structure of skeleton sequences.

To address this challenge, we propose Skeleton-Cache–a training-free test-time adaptation (TF-TTA) module that improves model generalizability to unseen actions during inference without requiring parameter updates or access to the training data. Skeleton-Cache builds and dynamically updates a non-parametric cache during testing and reformulates SZAR as a lightweight retrieval-based classification task. Instead of relying on a single holistic embedding–which may overlook subtle cues essential for distinguishing similar actions–Skeleton-Cache constructs a structured feature representation for both query and cache entries, composed of global descriptors and multiple fine-grained local descriptors that reflect key body regions and motion phases. This structured, multi-scale representation enables the model to capture subtle, discriminative patterns in skeletal sequences and allows the cache to store key patterns that remain consistent across different samples of the same class, leading to more precise feature matching and improved recognition of complex, previously unseen actions.

At test time, we compute the similarity between each descriptor of the query and the corresponding descriptors in the cache keys to obtain descriptor-wise class logits, which are then fused to obtain the final class prediction. A straightforward fusion approach would be to average these logits; however, such uniform treatment ignores the fact that different descriptors may vary in importance depending on the specific action. To address this, we leverage the strong semantic reasoning capabilities of large language models (LLMs) to provide linguistic priors in the form of importance weights for each descriptor. These weights encode high-level knowledge about the relevance of the global context, specific body regions, and motion phases for recognizing a given action–for example, recognizing the action "kicking" relies more on leg movements and cues from the final temporal phase. By incorporating these LLM-derived weights, our model adaptively emphasizes the most informative cues for each class, all without introducing any additional training.

Skeleton-Cache is a plug-and-play, training-free module that can be integrated into a wide range of SZAR backbones–including PURLS [33], SA-DVAE [15], SMIE [32] and SynSE [8]–offering dynamic and efficient test-time adaptation. Importantly, the cache evolves continuously during inference: features with high confidence are incorporated into the memory, while new or distinctive features create new entries. This non-parametric update mechanism supports on-the-fly adaptation to novel inputs without needing gradients or data augmentation.

Our contributions are summarized as follows:

- We introduce Skeleton-Cache, the first *training-free* test-time adaptation (TTA) framework for skeleton-based zero-shot action recognition (SZAR), enhancing the generalization of existing SZAR models to unseen actions without additional training.

- We design a structured cache within Skeleton-Cache that captures both a global descriptor and fine-grained spatial and temporal descriptors of skeleton sequences, enabling the model to retrieve subtle and discriminative cues critical for recognizing unseen actions more accurately.

- We leverage the powerful semantic reasoning capabilities of LLMs to derive class-specific importance weights, which guide an adaptive fusion of descriptor-wise predictions in a lightweight manner without introducing extra training.

- We conduct extensive experiments on NTU RGB+D 60/120 [20, 16] and PKU-MMD II[4], showing that Skeleton-Cache consistently boosts the performance of diverse SZAR backbones, with better results under both zero-shot and generalized zero-shot settings.

## 2 Related Work

### 2.1 Skeleton-based Zero-Shot Action Recognition

Skeleton ZSL has evolved rapidly from early metric-learning approaches by Wray *et al.* [28], who projected skeleton features into DeViSE-style semantic spaces, to more sophisticated methods like SynSE-ZSL [8] with syntactically guided pseudo samples for verb-level transfer, and SMIE [32] which maximized mutual information between visual and textual distributions for temporal cues. Recent advances leverage language models, with PURLS [33] aligning part-aware skeleton patterns with word embeddings and STAR [3] refining cross-modal alignment through dual prompts and side-information constraints, while generative approaches like SA-DVAE [15] synthesize unseen features to close the semantic gap. However, these methods lack mechanisms to adapt at inference, making them vulnerable to distribution shifts between seen and unseen classes. These shifts can occur even within the same dataset due to novel compositions of motion patterns, temporal variations, or underrepresented body part interactions. Without test-time adaptation, models are unable to recalibrate predictions in response to such discrepancies, limiting their effectiveness in real-world zero-shot scenarios. Our Skeleton-Cache addresses this by introducing the first *training-free* test-time adaptation module for SZAR–a plug-and-play solution that enhances diverse SZAR backbones without modifying learned parameters.

### 2.2 Test-Time Adaptation for Zero-Shot Learning

Test-Time Adaptation (TTA) research has progressed along two primary directions. Gradient-based methods update model parameters during inference, including Tent [26] which fine-tunes batch-norm statistics to minimize prediction entropy, TPT [24] which optimizes textual prompts via augmented views, PromptAlign [18] which aligns prompt token statistics with source data distributions, and DiffTPT [6] which employs diffusion models for diverse synthetic views—all suffering from latency and overfitting risks on smaller SZAR datasets due to their reliance on back-propagation and multiple forward passes. In contrast, Training-free alternatives offer higher efficiency, such as T3A [10] which dynamically re-weights class prototypes based on prediction confidence, CALIP [7] introduced a parameter-free attention module for cross-modal features, AdaNPC [31] which mixes test and source features to mitigate forgetting, and TDA [11] which constructs positive and negative feature caches for adapting models like CLIP. However, these methods are primarily designed for image classification and treat features holistically, overlooking the structured and fine-grained spatial-temporal information essential for distinguishing similar skeletal actions (e.g., *salute* vs. *wave*). Our Skeleton-Cache addresses these limitations with a design tailored specifically for SZAR. It avoids gradient-based updates entirely, mitigating overfitting and latency concerns. Besides, it models fine-grained information by decomposing skeleton inputs into structured global, spatial, and temporal descriptors. Furthermore, Skeleton-Cache uniquely leverages the semantic reasoning abilities of LLMs to guide descriptor fusion, enabling robust generalization to complex, unseen actions while remaining lightweight and plug-and-play.

## 3 Methodology

This section presents the full *Skeleton-Cache* pipeline, beginning with key preliminaries, followed by a detailed explanation of the proposed method

### 3.1 Preliminaries and Problem Definition

**Zero-shot Learning (ZSL)** [17] operates under specific conditions. Given a labeled dataset $\mathcal{D}_s = \{(\mathbf{x}_i, y_i)\}_{i=1}^{N_s}$, where $\mathbf{x}_i \in \mathcal{X}$ represents input features and $y_i \in \mathcal{Y}_s$ denotes seen class labels, the model is trained solely on seen classes and must handle unseen class samples during testing. In ZSL, the model is evaluated only on an unseen test set $\mathcal{D}_u = \{(\mathbf{x}_j, y_j)\}_{j=1}^{N_u}$, where $y_j \in \mathcal{Y}_u$ and

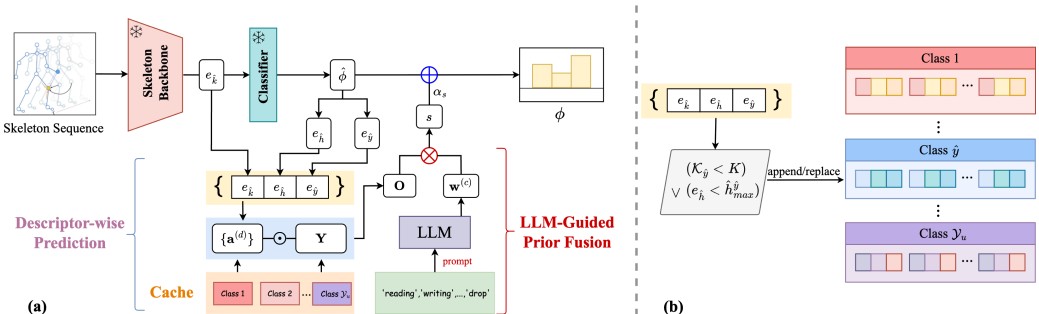

Figure 1: (a) Overview of the proposed pipeline that integrates our Skeleton-Cache with a frozen skeleton-based zero-shot action recognition (SZAR) model. (b) Illustration of the cache update process. $\mathcal{K}_{\hat{y}}$: entry count for class $\hat{y}$. K: entry limit per class in cache. $\hat{h}^{\hat{y}}_{max}$: The maximum value of $e_{\hat{h}}$ within $\hat{y}$ class block.

$\mathcal{Y}_s \cap \mathcal{Y}_u = \emptyset$. The prediction space is restricted to unseen labels: $f : \mathcal{X} \to \mathcal{Y}_u$. In Generalized ZSL (GZSL), the model must recognize samples from both seen and unseen classes, expanding the prediction space to $\mathcal{Y}_s \cup \mathcal{Y}_u$.

**Test-Time Adaptation (TTA).** To combat distribution shift during deployment, TTA updates the inference pipeline on-the-fly. *Gradient-based* TTA adjusts model parameters $\theta$ for every online mini-batch $\mathcal{B}_t = \{\mathbf{x}_i\}_{i=1}^B$ through one or a few optimisation steps:

$$\theta_t = \theta_{t-1} - \eta\nabla_\theta\mathcal{L}\big(\mathcal{B}_t; \theta_{t-1}\big), \tag{1}$$

where $\eta$ is the learning rate and $\mathcal{L}$ an adaptation objective. By contrast, *training-free* TTA (TF-TTA) freezes $\theta$ and updates only a light-weight auxiliary state $\mathcal{S}$–for example, a cache or running statistics [10, 11]. Given frozen features $\mathbf{z}_i = \Phi(\mathbf{x}_i)$ and their zero-shot predictions $\hat{\mathbf{y}}_i$, TF-TTA performs

$$\mathcal{S}_t = \mathcal{U}\big(\mathcal{S}_{t-1}, \{\mathbf{z}_i, \hat{\mathbf{y}}_i\}_{i=1}^B\big), \tag{2}$$

where $\mathcal{U}$ denotes the update function that modifies the auxiliary state based on current batch information, achieving real-time throughput with *zero* gradient cost. Our Skeleton-Cache instantiates $\mathcal{S}$ as the cache detailed below.

## 3.2 Skeleton-Cache

Current methods for skeleton-based zero-shot action recognition (SZAR) are trained solely on seen classes and lack the ability to adapt dynamically at inference time, which can limit their generalization to unseen actions. To address this, we present *Skeleton-Cache*, a training-free test-time adaptation module that can be integrated into SZAR pipelines for improved generalization to unseen actions without requiring access to training data or any model updates.

Specifically, Skeleton-Cache reframes action inference as a dynamic retrieval process. As illustrated in Fig. 1, it constructs and continuously updates a non-parametric cache during test time. For each incoming test sequence, the system computes descriptor-wise predictions and then refines the final classification by leveraging linguistic priors derived from large language models (LLMs). Below we describe the details.

**Cache Construction and Update:** The cache is structured as class-specific blocks, one for each action class, with each block holding up to $K$ entries. Each entry is a tuple $(\mathbf{e}_{\hat{k}}, \mathbf{e}_{\hat{y}}, \mathbf{e}_{\hat{h}})$ that captures essential information for retrieval and update: (i) $\mathbf{e}_{\hat{k}}$ is the feature representation (cache key) of a skeleton sequence, used to compute similarity with incoming queries. (ii) $\mathbf{e}_{\hat{y}}$ is the predicted label of the sequence (cache value), used in computing the final prediction for the query. (iii) $\mathbf{e}_{\hat{h}}$ is the prediction confidence associated with the sample, used to assess the reliability of the sample and guide cache updates. The cache is initialized as empty. As test samples arrive, we selectively insert high-confidence entries into their corresponding class-specific blocks. This maintains a compact yet representative memory for inference. Below, we describe how to extract each tuple and the strategy for updating the cache.

*Cache Key* $\mathbf{e}_{\hat{k}}$: We denote an incoming test skeleton sequence as $\mathbf{x} \in \mathbb{R}^{C \times T \times V \times M}$, where $C$ is the channel dimension, $T$ is the number of frames, $V$ is the number of joints, and $M$ is the number of

persons. Given a trained and frozen SZAR model with a skeleton feature encoder (e.g., ST-GCN [30]), we extract features for each skeleton sequence and average over persons to obtain a latent tensor $\mathbf{F} \in \mathbb{R}^{N \times T \times V}$, where $N$ is the feature dimension after encoding. To obtain the cache key, one straightforward approach is to apply global pooling over $\mathbf{F}$ to produce a single representation. However, relying solely on global context can be limiting for action recognition. While some actions are distinguishable at a coarse level, others differ in subtle, local patterns that holistic features may overlook. For instance, "wave" and "salute" may appear similar globally but differ in fine-grained cues like hand posture or local motion dynamics. In addition, key local patterns often remain consistent across samples of the same class, making them more reliable for retrieval and generalization. To improve the model's ability to distinguish between similar actions and accurately recognize intra-class instances, we compute both global and local spatial temporal descriptors as the cache key:

$$\mathbf{s}_p = \frac{1}{|\mathcal{V}_p|T} \sum_{v \in \mathcal{V}_p} \sum_{t=1}^{T} \mathbf{F}_{:,t,v}, \quad \mathbf{t}_z = \frac{1}{V|\mathcal{T}_z|} \sum_{t \in \mathcal{T}_z} \sum_{v=1}^{V} \mathbf{F}_{:,t,v}, \quad \mathbf{g} = \frac{1}{VT} \sum_{t=1}^{T} \sum_{v=1}^{V} \mathbf{F}_{:,t,v}, \tag{3}$$

where $\mathbf{s}_p \in \mathbb{R}^N$, $\mathbf{t}_z \in \mathbb{R}^N$ and $\mathbf{g} \in \mathbb{R}^N$ denote the local spatial, local temporal and global descriptors, respectively. $\{\mathcal{V}_p\}_{p=1}^{P}$ denotes $P$ predefined joint groups corresponding to body parts (e.g., head, torso, arms, legs). $\{\mathcal{T}_z\}_{z=1}^{Z}$ represents $Z$ predefined temporal segments (e.g., beginning, middle, end) that together span the entire sequence. The final cache key $\mathbf{e}_{\hat{k}} \in \mathbb{R}^{(P+Z+1) \times N}$ is formed by first expanding each descriptor with an additional dimension to form row vectors in $\mathbb{R}^{1 \times N}$ and concatenating them along the first dimension to form a matrix:

$$\mathbf{e}_{\hat{k}} = \text{concat}\left(\mathbf{g}, \mathbf{s}_1, \ldots, \mathbf{s}_P, \mathbf{t}_1, \ldots, \mathbf{t}_Z\right) \in \mathbb{R}^{(P+Z+1) \times N}. \tag{4}$$

*Cache Value $\mathbf{e}_{\hat{y}}$ and confidence $\mathbf{e}_{\hat{h}}$*: As shown in Fig. 1(a), given a testing skeleton sequence, we pass it through a trained and fixed SZAR model to obtain prediction logits $\hat{\phi}$. Applying the softmax function yields the class probability distribution $\hat{\rho} = \text{softmax}(\hat{\phi})$ over the unseen classes $\mathcal{Y}_u$. The predicted label is $\hat{y} = \arg\max_j \hat{\rho}_j$ and is encoded as a one-hot vector $\mathbf{e}_{\hat{y}} \in \mathbb{R}^{|\mathcal{Y}_u|}$. The prediction confidence is quantified by the entropy of the distribution:

$$\mathbf{e}_{\hat{h}} = -\sum_{j=1}^{|\mathcal{Y}_u|} \hat{\rho}_j \log \hat{\rho}_j, \tag{5}$$

where lower entropy indicates higher confidence.

*Cache Update:* For each incoming test sequence, we extract its tuple $(\mathbf{e}_{\hat{k}}, \mathbf{e}_{\hat{h}}, \mathbf{e}_{\hat{y}})$. We then examine the cache block corresponding to the predicted class $\hat{y}$. As shown in Fig. 1(b), if the number of entries for class $\hat{y}$ ($\mathcal{K}_{\hat{y}}$) is below the maximum capacity $K$, the tuple is directly appended. Otherwise, we identify the entry within class $\hat{y}$ that has the highest entropy $\hat{h}_{max}^{\hat{y}}$ (i.e., the lowest confidence). If the current test sample's entropy is lower (indicating higher confidence), it replaces the least confident entry with $\hat{h}_{max}^{\hat{y}}$; otherwise, no update is made.

**Cache Retrieval for Descriptor-wise Prediction:** For each test sample, we compute its local spatial, local temporal, and global features as defined in Eq. 3, forming a descriptor matrix $\mathbf{e_q} = \left[\mathbf{q}^{(0)}, \mathbf{q}^{(1)}, \ldots, \mathbf{q}^{(P+Z)}\right] \in \mathbb{R}^{(P+Z+1) \times N}$, where $\mathbf{q}^{(0)}$ is the global descriptor, $\{\mathbf{q}^{(1)}, \cdots, \mathbf{q}^{(P)}\}$ are the local spatial descriptors, and $\{\mathbf{q}^{(P+1)}, \cdots, \mathbf{q}^{(P+Z)}\}$ are the local temporal descriptors. Once the cache is populated, each query descriptor is compared with the corresponding descriptor of all cached entries to compute similarity scores. These similarity scores are used to weight the cached values (one-hot labels), producing a descriptor-wise prediction vector for each descriptor type. The set of descriptor-wise predictions is then fused to generate the final class prediction.

Specifically, for each descriptor type $d \in \{0, 1, \ldots, P + Z\}$, we compare the corresponding query vector $\mathbf{q}^{(d)}$ with all cached keys of the same descriptor type. Let $\mathbf{k}_{j,i}^{(d)}$ represent the cached key of type $d$ from the $i$-th entry belonging to class $j$, where $j \in \{1, 2, \ldots, |\mathcal{Y}_u|\}$ and $i \in \{1, 2, \ldots, K\}$. The similarity, or affinity, between the query descriptor and a cached key is calculated by:

$$a_{j,i}^{(d)} = \exp\left[-\beta\left(1 - \cos(\mathbf{q}^{(d)}, \mathbf{k}_{j,i}^{(d)})\right)\right], \tag{6}$$

where $\beta > 0$ is a temperature parameter controlling the sharpness of the similarity distribution, and $\cos(\cdot, \cdot)$ denotes the cosine similarity.

All affinity scores computed across the cached entries of the $|\mathcal{Y}_u|$ classes are concatenated into a single row vector. Assuming that each class has exactly $K$ entries (resulting in a total of $|\mathcal{Y}_u| \times K$ entries), we define the affinity vector for descriptor type $d$ $\mathbf{a}^{(d)} \in \mathbb{R}^{1 \times (|\mathcal{Y}_u| \times K)}$ as:

$$\mathbf{a}^{(d)} = [a_{1,1}^{(d)}, \ldots, a_{1,K}^{(d)}, a_{2,1}^{(d)}, \ldots, a_{2,K}^{(d)}, \ldots, a_{|\mathcal{Y}_u|,1}^{(d)}, \ldots, a_{|\mathcal{Y}_u|,K}^{(d)}]. \tag{7}$$

We also construct a label matrix $\mathbf{Y} \in \mathbb{R}^{(|\mathcal{Y}_u| \times K) \times |\mathcal{Y}_u|}$ from all entries, where each row is a cache value, i.e., a one-hot vector indicating the class label of an entry. The descriptor-wise prediction is then computed as:

$$\mathbf{o}^{(d)} = \mathbf{a}^{(d)} \mathbf{Y} \in \mathbb{R}^{1 \times |\mathcal{Y}_u|}, \tag{8}$$

where $\mathbf{o}^{(d)}$ represents the retrieved class logits derived from the similarity scores associated with descriptor type $d$. Although we assume a fixed number of $K$ entries per class for clarity, the above formulas naturally extend to cases where the number of cached entries varies across classes.

We repeat the above process for all $P + Z + 1$ descriptor types, resulting in a collection of descriptor-wise prediction vectors $\{\mathbf{o}^{(d)}\}_{d=0}^{P+Z}$. Each vector $\mathbf{o}^{(d)}$ offers a complementary class prediction based on different types of descriptors–global, spatial, or temporal. These predictions are subsequently fused using the LLM-guided weighting scheme described below.

It is worth mentioning that the above computation is highly efficient, as it primarily involves matrix multiplications with sparse one-hot label matrices and vectors of maximum length $|\mathcal{Y}_u| \times K$. The computation can be naturally parallelized, further enhancing efficiency. Additionally, the number of descriptors is constant (independent of the skeleton sequence length), meaning that the computational complexity remains unaffected by the sequence duration. This enables Skeleton-Cache to operate in real-time, even for high-frame-rate streams.

**LLM-Guided Prior for Fusing Descriptor-wise Predictions:** To aggregate the descriptor-wise predictions $\{\mathbf{o}^{(d)}\}_{d=0}^{P+Z}$ into the final class logits for a query sequence, we account for the fact that different descriptor type–global, spatial, or temporal–may vary in relevance depending on the specific action. Rather than using uniform averaging, we leverage a large language model (LLM) to derive a prior in the form of class-specific weights that capture the relative importance of each descriptor. LLMs are powerful general-purpose reasoning engines with rich semantic knowledge and contextual understanding, making them well-suited for estimating informative priors in a training-free manner. These weights are then used to guide a weighted fusion of the descriptor-wise predictions, enabling a more informed and adaptive final classification. Below we describe the details.

*Prompt Design and Weight Extraction.* For each action category $c$ in $\mathcal{Y}_u$, we issue a single prompt to the LLM to obtain: (i) importance scores for the $P$ spatial body regions, normalized to sum to 1, (ii) importance scores for the $Z$ temporal phases, normalized to sum to 1, and (iii) a relative preference score $\gamma \in [0, 1]$ balancing global versus local descriptors.

Our prompt captures three key aspects: which body parts are most distinctive (e.g., arms for "waving"), which temporal phases are critical (e.g., the apex of "jumping"), and whether the action is better recognized holistically or through local components.

The LLM directly provides all necessary components in its response: spatial scores $\{w_{\text{spa}}^{(p)}\}_{p=1}^{P}$ quantifying body part relevance, temporal scores $\{w_{\text{tmp}}^{(z)}\}_{z=1}^{Z}$ reflecting the importance of temporal segments, and a global-local preference $\gamma \in [0, 1]$ indicating whether the action is better represented as a holistic motion pattern or as a composition of spatial-temporal primitives. These raw outputs are combined into a $(P + Z + 1)$-dimensional importance vector as follows:

$$\tilde{\mathbf{w}}^{(c)} = \left[\gamma, \, (1 - \gamma)\, w_{\text{spa}}^{(1)}, \ldots, (1 - \gamma)\, w_{\text{spa}}^{(P)}, \, (1 - \gamma)\, w_{\text{tmp}}^{(1)}, \ldots, (1 - \gamma)\, w_{\text{tmp}}^{(Z)}\right] \tag{9}$$

which is subsequently $\ell_1$-normalized to obtain the final weight vector $\mathbf{w}^{(c)}$. All components–$w_{\text{spa}}^{(p)}$, $w_{\text{tmp}}^{(z)}$, and $\gamma$–are derived from the LLM's feedback, effectively translating its semantic understanding of the action into computational attention over descriptors. Details on the prompt design, implementation, and analysis of the LLM responses are provided in the Appendix.

*Weighted fusion of Descriptor-wise Predictions:* Given the set of descriptor-wise prediction vectors $\{\mathbf{o}^{(d)}\}_{d=0}^{P+Z}$, where each $\mathbf{o}^{(d)} \in \mathbb{R}^{1\times|\mathcal{Y}_u|}$ corresponds to the class logit vector of a descriptor, we stack them along the first dimension to form a matrix:

$$\mathbf{O} = \text{concat}\left(\mathbf{o}^{(0)}, \mathbf{o}^{(1)}, \dots, \mathbf{o}^{(P+Z)}\right) \in \mathbb{R}^{(P+Z+1)\times|\mathcal{Y}_u|}, \tag{10}$$

where each row corresponds to the prediction logits from a specific descriptor. Specifically, $\mathbf{o}^{(0)}$ is the prediction from the global descriptor, $\{\mathbf{o}^{(p)}\}_{p=1}^{P}$ are the predictions from the local spatial descriptors, and $\{\mathbf{o}^{(P+z)}\}_{z=1}^{Z}$ are the predictions from the temporal descriptors.

We then apply the LLM-derived class-specific weight vector $\mathbf{w}^{(c)}$ to $\mathbf{O}$ to perform a weighted fusion of the descriptor-wise predictions:

$$\mathbf{s} = \mathbf{w}^{(c)}\mathbf{O} \in \mathbb{R}^{1\times|\mathcal{Y}_u|}. \tag{11}$$

This weighted fusion mechanism ensures that descriptors most relevant to a given action class–based on the LLM's semantic prior–receive greater emphasis in the final prediction. By leveraging linguistic priors to guide the combination of descriptor-wise logits, this approach enhances prediction accuracy without introducing any trainable parameters, thereby preserving the training-free nature of Skeleton-Cache.

For each testing sample, we leverage the fused logits $\mathbf{s}$ to enhance the original zero-shot logit $\hat{\phi}$ produced by the frozen SZAR model by:

$$\phi = \hat{\phi} + \alpha_s\,\mathbf{s}, \tag{12}$$

where $\alpha_s$ is a balancing coefficient. The posterior used for final prediction is $\rho = \text{softmax}(\phi)$.

## 4   Experimental and Results

### 4.1   Datasets

**NTU RGB+D 60 Dataset** [20]. It contains 56,880 skeleton sequence samples of 60 actions, with 40 individual subjects captured from 80 distinct camera viewpoints. Each sample provides a temporal sequence of the 3-D location coordinates for 25 human body joints per performer. The maximum performer number is 2, and the coordinate values are padded as 0 when the corresponding performer is unavailable (e.g., single-person actions).

**NTU RGB+D 120 Dataset** [16]. It is the extension of the NTU RGB+D 60, contains 114,480 samples for 120 actions performed by 106 individual subjects captured from 155 distinct camera viewpoints.

**PKU-MMD Dataset** [4]. It contains two phases for action recognition with increasing difficulty, which covers the same multi-modalities as the NTU dataset. The actions are collected into 51 action categories, and almost 20000 instances are included.

### 4.2   Descriptor Partitioning and LLM-Guided Weight Generation

To construct the structured cache keys, we partition skeleton sequences along both spatial and temporal dimensions, enabling Skeleton-Cache to capture fine-grained motion patterns efficiently.

For spatial partitioning, we divide the skeleton into four semantically meaningful body regions based on the Kinect v2 25-joint model: head (joints 2, 3, 4, 8, 20), torso (joints 0, 1, 4, 8, 12, 16, 20), arms (joints 4–11, 21–24), and feet (joints 0, 12–19). These groupings align with natural anatomical boundaries and capture localized motion patterns relevant to different action types. For temporal partitioning, we uniformly segment each sequence of length $T$ into three phases—beginning ($t \in [1, \lfloor T/3 \rfloor]$), middle ($t \in [\lfloor T/3 \rfloor + 1, \lfloor 2T/3 \rfloor]$), and end ($t \in [\lfloor 2T/3 \rfloor + 1, T]$)—without requiring manual annotation of action phases. The spatial features $s_p$ and temporal features $t_z$ are then computed by averaging feature representations across the corresponding joints and frames as defined in Eq. 3. Complete joint assignments are detailed in the Appendix.

For the LLM-guided weighting mechanism, we leverage GPT-4o to generate class-specific semantic priors in a training-free manner. For each action class, we issue a single structured prompt requesting three components: normalized spatial importance scores across the four body regions, normalized temporal importance scores across the three phases, and a global-local preference parameter $\gamma \in [0, 1]$ indicating whether the action is better recognized holistically or through local components. The LLM directly returns these values, which we combine via Eq. 9 to form the final weight vector $\mathbf{w}^{(c)}$ without any manual tuning or post-processing. This design preserves the training-free nature of our approach. The complete prompt template is provided in the Appendix.

**LLM-Guided Weight Generation.** The LLM serves as a source of structured semantic knowledge, generating class-specific priors in a completely training-free manner. To obtain weights for spatial importance, temporal importance, and global-vs-local preference, we query GPT-4o once per action class using a structured prompt without any manual adjustment or post-processing. The LLM directly returns normalized spatial weights $\{w_{\text{spa}}^{(p)}\}_{p=1}^{P}$, temporal weights $\{w_{\text{tmp}}^{(z)}\}_{z=1}^{Z}$, and a global-local preference parameter $\gamma \in [0, 1]$, which are combined according to Eq. 9. We deliberately avoid fine-tuning these weights to preserve the training-free nature of our approach. The quantitative results (Table 3) and qualitative visualizations (Appendix D.1) empirically validate that LLM-generated weights align well with human intuition and significantly outperform uniform averaging, demonstrating the effectiveness of incorporating semantic priors for test-time adaptation in zero-shot settings. The complete prompt template and example responses are provided in Appendix B.2.

## 4.3 Implementation Details

To evaluate the generalization ability of our method, we follow the training and testing protocols described in SynSE [8], SMIE [32], PURLS [33], and SA-DVAE [15]. For every baseline and for our *Skeleton-Cache*, the experimental settings strictly match those reported in the original papers. During inference all pre-trained weights are loaded once and kept frozen. The cache size is fixed to $K = 8$ prototypes per unseen class; the body-part granularity is $P = 4$ (head, torso, arms, legs) and the temporal segmentation is $Z = 3$ (begin, middle, end). We use a test batch size of 1 to emulate streaming deployment. To construct the *LLM-Guided Prior Matrix*, we query GPT-4o via the API framework. All experiments run on a single NVIDIA RTX 4090 GPU. Additional implementation details are provided in the **Appendix**.

As described in the Methods section, we conducted our experiments under both zero-shot learning and generalized zero-shot learning settings. For zero-shot learning, we use Top-1 recognition accuracy on unseen test samples $\mathcal{D}_{te}^{u}$ as our metric. For generalized zero-shot learning, we use the accuracy on both seen test samples $\mathcal{D}_{te}^{u}$ and unseen test samples $\mathcal{D}_{te}^{u}$, as well as their harmonic mean as metrics.

Our method follows the settings of existing baseline methods, adopting the split settings as [8], with all model training conducted on the seen train dataset. Due to fixed splits potentially not reflecting method performance well, recent papers (SMIE [32], SA-DVAE [15], etc.) have established a new setting where they implement three different random class splits of 55/5, 110/10, and 46/5 for NTU60, NTU-120, and PKU-MMD datasets respectively. Our paper has also completed this baseline, with results shown in Table 2.

## 4.4 Comparison with State-of-the-Art

Table 1 contrasts *Skeleton-Cache* with leading zero-shot skeleton action recognition approaches on NTU RGB+D 60 and 120, while Table 2 reports results under the random-split protocol and on PKU-MMD, underscoring generalization capabilities. Attaching the cache to heterogeneous backbones yields uniform accuracy improvements of 4–8 percentage points across all datasets and splits. Notable results include SA-DVAE climbing from 82.37% to 89.41% (+7.04 pp) on NTU60 55/5, PURLS reaching 85.46% (+6.24 pp) on the same split, and similar gains on the more challenging NTU120 96/24 split. The SA-DVAE + Skeleton-Cache combination achieves 89.86% ZSL accuracy on NTU60 and 71.05% on PKU-MMD, surpassing all published numbers. These improvements extend to the GZSL setting with harmonic mean increases of 4–7 pp, confirming benefits for both seen and unseen classes. Across varied evaluation settings, Skeleton-Cache consistently delivers state-of-the-art performance while adding minimal memory overhead and latency, demonstrating practicality for real-time deployments.

Table 1: Comparison on NTU RGB+D datasets. **ZSL**: zero-shot Top-1 accuracy. **S/U/H**: seen, unseen, and harmonic mean in GZSL. **SC**: Skeleton-Cache, our proposed method.

| | NTU RGB+D 60 | | | | | | | | NTU RGB+D 120 | | | | | | | |
| | 55/5 Split | | | | 48/12 Split | | | | 110/10 Split | | | | 96/24 Split | | | |
| Method | ZSL | S | U | H | ZSL | S | U | H | ZSL | S | U | H | ZSL | S | U | H |
|---|---|---|---|---|---|---|---|---|---|---|---|---|---|---|---|---|
| ReViSE [9] | 53.91 | 74.22 | 34.73 | 47.32 | 17.49 | 62.36 | 20.77 | 31.16 | 55.04 | 48.69 | 44.84 | 46.68 | 32.38 | 49.66 | 25.06 | 33.31 |
| JPoSE [28] | 64.82 | 64.44 | 50.29 | 56.49 | 28.75 | 60.49 | 20.62 | 30.75 | 51.93 | 47.66 | 46.40 | 47.05 | 32.44 | 38.62 | 22.79 | 28.67 |
| CADA-VAE [19] | 76.84 | 69.38 | 61.79 | 65.37 | 28.96 | 51.32 | 27.03 | 35.41 | 59.53 | 47.16 | 49.78 | 48.44 | 35.77 | 41.11 | 34.14 | 37.31 |
| SynSE [8] | 75.81 | 61.27 | 56.93 | 59.02 | 33.30 | 52.21 | 27.85 | 36.33 | 62.69 | 52.51 | 57.60 | 54.94 | 38.70 | 56.39 | 32.25 | 41.04 |
| SMIE [32] | 77.98 | – | – | – | 40.18 | – | – | – | 65.74 | – | – | – | 45.30 | – | – | – |
| PURLS [33] | 79.22 | – | – | – | 40.99 | – | – | – | 71.95 | – | – | – | 52.01 | – | – | – |
| SA-DVAE [15] | 82.37 | 62.28 | 70.80 | 66.27 | 41.38 | 50.20 | 36.94 | 42.56 | 68.77 | 61.10 | 59.75 | 60.42 | 46.12 | 58.82 | 35.79 | 44.50 |
| SynSE+SC | 79.73↑3.92 | 62.31 | 65.37 | 63.80↑4.78 | 38.82↑5.52 | 54.02 | 34.88 | 42.39↑6.06 | 68.47↑5.78 | 53.44 | 64.80 | 58.57↑3.63 | 44.10↑5.40 | 58.28 | 37.86 | 45.90↑4.86 |
| SMIE+SC | 82.63↑4.65 | – | – | – | 44.17↑3.99 | – | – | – | 72.98↑7.24 | – | – | – | 50.44↑5.14 | – | – | – |
| PURLS+SC | 85.46↑6.24 | – | – | – | 45.22↑4.23 | – | – | – | 77.60↑5.65 | – | – | – | 56.83↑4.82 | – | – | – |
| SA-DVAE+SC | 89.41↑7.04 | 64.62 | 79.16 | 71.15↑4.88 | 47.83↑6.45 | 53.22 | 44.41 | 48.42↑5.86 | 74.29↑5.52 | 62.19 | 65.83 | 63.96↑3.54 | 53.14↑7.02 | 59.94 | 44.13 | 50.83↑6.33 |

Table 2: Random-split comparison on three datasets. **ZSL**: Top-1 zero-shot accuracy; **GZSL**: harmonic mean for GZSL; **SC**: Skeleton-Cache, our proposed method.

| | NTU 60 | | NTU 120 | | PKU-MMD | |
| Method | ZSL | GZSL | ZSL | GZSL | ZSL | GZSL |
|---|---|---|---|---|---|---|
| ReViSE [9] | 60.94 | 60.34 | 44.90 | 40.34 | 59.34 | 49.82 |
| JPoSE [28] | 59.44 | 60.05 | 46.69 | 43.69 | 57.17 | 51.64 |
| CADA-VAE [19] | 61.84 | 66.38 | 45.15 | 45.64 | 60.74 | 45.75 |
| SynSE [8] | 64.19 | 67.47 | 47.28 | 43.47 | 53.85 | 49.47 |
| SMIE [32] | 65.08 | – | 46.40 | – | 60.83 | – |
| SA-DVAE [15] | 84.20 | 75.27 | 50.67 | 47.54 | 66.54 | 54.72 |
| SynSE+SC | 68.22↑4.03 | 73.24↑5.77 | 51.63↑4.35 | 50.08↑6.61 | 57.63↑3.78 | 55.31↑5.84 |
| SMIE+SC | 71.24↑6.16 | – | 50.85↑4.45 | – | 66.76↑5.93 | – |
| SA-DVAE+SC | 89.86↑5.66 | 80.21↑4.94 | 56.18↑5.51 | 51.94↑4.40 | 71.05↑4.51 | 58.49↑3.77 |

## 4.5 Ablation Studies & Qualitative Analysis

**Hyperparameter Analysis.**

Figure 2 illustrates the impact of three key hyperparameters. For cache size $K$ (Fig. 2a), performance improves as we increase from 2 to 8 entries per class, with diminishing returns beyond that point, indicating that a moderate cache effectively captures class-specific patterns. The balancing coefficient $\alpha_s$ (Fig. 2b) shows optimal performance around 5.0, balancing the influence of cache-retrieved logits against original zero-shot predictions. The temperature parameter $\beta$ (Fig. 2c) reaches peak performance

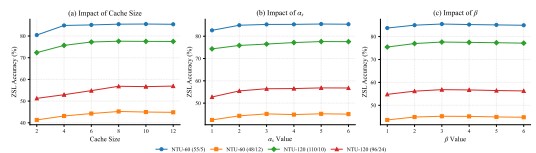

Figure 2: Ablation studies on key hyperparameters of the Skeleton-Cache mechanism. (a) Cache size $K$. (b) Balancing coefficient $\alpha_s$. (c) Similarity temperature parameter $\beta$.

at 3.0, providing the ideal sharpness for the similarity distribution used in descriptor comparison. Across all settings, performance differences are relatively small and trends are stable across different data splits, indicating that Skeleton-Cache is robust to hyperparameter choices.

**Weighting strategy.** To show the benefits of using LLM prior for fusing descriptor-wise predictions, we conduct two other baselines: **Random weights:** weights are randomly generated for each descriptor, **Uniform weights:** the same weight is assigned to all descriptors. The comparison results are shown in Table 3 (top). Random weights degrade accuracy because they over-emphasise noisy parts, whereas uniform weights yield modest gains by averaging all cues. LLM-guided weights provide the largest and most consistent improvements, confirming that class-specific priors help the cache focus on informative body regions and motion phases. The class-specific weight generated by the LLM for example actions are provided in the Appendix.

**Descriptor granularity.** We examine the contribution of different descriptor types in Table 3 (bottom). Using only the global descriptor (**g**) already surpasses the baseline, but adding spatial ($\{\mathbf{s}_p\}_{p=1}^{P}$) or temporal ($\{\mathbf{t}_z\}_{z=1}^{Z}$) descriptors is even more beneficial. Spatial information is slightly more valuable than temporal, suggesting that body part localization provides stronger cues for zero-shot generalization. Yet the full eight-part design that combines global, spatial, and temporal

Table 3: Ablation study on *Skeleton-Cache* components. **Weighting**: Different fusion strategies for 8-part descriptors; **Granularity**: Comparison of descriptor types; Values show ZSL accuracy (%); ↑ x indicates gain over baseline.

| Method | NTU Dataset Splits | | | |
| --- | --- | --- | --- | --- |
| | 55/5 (NTU60) | 48/12 (NTU60) | 110/10 (NTU120) | 96/24 (NTU120) |
| PURLS baseline | 79.22 | 40.99 | 71.95 | 52.01 |
| *Weighting strategy (full descriptors)* | | | | |
| Random weights | 77.31 ↓1.91 | 38.56 ↓2.43 | 68.02 ↓3.93 | 46.47 ↓5.54 |
| Uniform weights | 83.67 ↑4.45 | 44.88 ↑3.89 | 75.54 ↑3.59 | 54.37 ↑2.36 |
| LLM weights | **85.46**↑6.24 | **45.22**↑4.23 | **77.60**↑5.65 | **56.83**↑4.82 |
| *Descriptor granularity (LLM weights fixed)* | | | | |
| Global-only | 82.60 ↑3.38 | 42.13 ↑1.14 | 73.08 ↑1.13 | 53.57 ↑1.56 |
| Spatial-only (4 parts) | 83.79 ↑4.57 | 44.68 ↑3.69 | 75.46 ↑3.51 | 56.07 ↑4.06 |
| Temporal-only (3 parts) | 82.14 ↑2.92 | 43.32 ↑2.33 | 74.52 ↑2.57 | 54.41 ↑2.40 |
| Full (8 parts) | **85.46**↑6.24 | **45.22**↑4.23 | **77.60**↑5.65 | **56.83**↑4.82 |

Table 4: Comparison with other TTA methods on NTU-RGBD 60/120 under different SZAR splits. Throughput is measured in samples/ms.

| Method | NTU-RGBD 60 | | NTU-RGBD 120 | | Throughput |
| --- | --- | --- | --- | --- | --- |
| | 55/5 | 48/12 | 110/10 | 96/24 | |
| PURLS baseline | 79.22 | 40.99 | 71.95 | 52.01 | 10.09 |
| TPT [24] | 40.11 | 27.98 | 22.10 | 17.10 | 0.70 |
| DiffTPT [6] | 42.27 | 28.34 | 23.43 | 20.01 | 0.59 |
| CALIP [7] | 80.02 | 45.10 | 72.27 | 51.84 | 2.48 |
| Ours (Skeleton-Cache) | **85.46** | **45.22** | **77.60** | **56.83** | 3.07 |

descriptors delivers the best performance on every split. The accuracy margin over PURLS reaches 6.24 percentage points on NTU60 55/5.

**Comparison with other TTA methods.** As shown in the table on the right, we experimented with different TTA methods on the PURLS baseline. TPT [24] and DiffTPT [6]perform poorly likely because they use learnable prompts that significantly alter CLIP's text features, disrupting the delicate alignment between skeleton and text features—a critical issue given the limited generalization capability of skeleton-based zero-shot models trained on small datasets. CALIP [7] enhances modality alignment directly at the feature level and achieves significant improvements. Our method further improves performance by introducing fine-grained descriptors and prior knowledge, which increases inter-class separability.

# 5 Conclusion

We present Skeleton-Cache, a fully training-free test-time adaptation framework for zero-shot skeleton action recognition. By encoding each input sequence into structured, fine-grained descriptors, reframing recognition as a retrieval task via a lightweight, non-parametric cache, and integrating retrieved scores through LLM-guided importance weights, our method enhances prediction accuracy on novel classes across diverse zero-shot action recognition models without any gradient updates.

**Limitation.** As a training-free method, our approach builds on the representations extracted by a pre-trained SZAR model. While this enables broad applicability and eliminates the need for additional training, the effectiveness of cache retrieval may vary depending on the quality of input skeletons. In challenging scenarios with severe occlusions or pose noise, performance may be moderately affected.

**Broader impacts.** By combining test-time adaptation with zero-shot skeleton recognition, Skeleton-Cache can empower adaptive gesture understanding in assistive robotics and real-time analytics in sports, while also raising privacy considerations that warrant ethical deployment and clear consent protocols.

## 6 Acknowledgment

This research was supported by the Australian Government through the Australian Research Council's DECRA funding scheme (Grant No.: DE250100030) and Discovery Projects scheme (Grant No: DP210101682).

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

# Appendix

This appendix provides detailed supplementary material to support the main paper, *Boosting Skeleton-based Zero-Shot Action Recognition with Training-Free Test-Time Adaptation.* It is organized into four sections: (1) **Algorithm** gives the full Skeleton-Cache pseudocode and clarifies notation. (2) **Implementation Details** covers the efficiency analysis, LLM prompt design, and the spatial/temporal partitioning strategy required for replication. (3) **More Quantitative Results** extends the empirical study with extra ZSL methods, comparisons with generic TTA baselines, and cache update with adapted logits. Finally, (4) **Analytical Visualizations** analyzes adaptation effects through weight heat-maps, confusion matrices, top-5 prediction changes, and per-class accuracy changes. These sections collectively enhance the understanding of Skeleton-Cache's functionality, reproducibility, and performance.

## A   Algorithm

This section provides a formal description of the Skeleton-Cache algorithm through pseudocode and clarifies the notation used throughout our paper. The algorithm is designed to enhance the generalization of skeleton-based zero-shot action recognition (SZAR) models by dynamically maintaining a non-parametric cache of confident exemplars and integrating similarity-based predictions with LLM-guided weights, all without requiring model retraining.

---

**Algorithm 1** Skeleton-Cache: Training-Free Test-Time Adaptation for SZAR

---

**Require:** Test skeleton sequences, action class descriptions, number of unseen classes $|\mathcal{Y}_u|$, cache size $K$, spatial groups $P$, temporal segments $Z$, hyperparameters $\alpha_s, \beta$
**Ensure:** Adapted class predictions $\hat{y}$

1: Initialize caches $\mathcal{C}_j \leftarrow \emptyset$ for $j = 1, \ldots, |\mathcal{Y}_u|$
2: **for all** test samples **do**                    ▷ Phase 1: Feature Extraction
3:     Extract skeleton features $F \in \mathbb{R}^{N \times T \times V}$ with $\mathcal{M}$
4:     $g \leftarrow \frac{1}{VT} \sum_{t=1}^{T} \sum_{v=1}^{V} F_{:,t,v}$
5:     **for** $p = 1$ **to** $P$ **do**
6:         $s_p \leftarrow \frac{1}{|V_p|T} \sum_{v \in V_p} \sum_{t=1}^{T} F_{:,t,v}$
7:     **for** $z = 1$ **to** $Z$ **do**
8:         $t_z \leftarrow \frac{1}{V|T_z|} \sum_{t \in T_z} \sum_{v=1}^{V} F_{:,t,v}$
9:     $e_k \leftarrow \text{concat}(g, s_1, \ldots, s_P, t_1, \ldots, t_Z)$       ▷ Phase 2: Initial Prediction
10:     Obtain initial label $\hat{y}$ and entropy $\hat{h}$ from $\mathcal{M}$       ▷ Phase 3: Cache Update
11:     **if** $|\mathcal{C}_{\hat{y}}| < K$ **then**
12:         $\mathcal{C}_{\hat{y}} \leftarrow \mathcal{C}_{\hat{y}} \cup \{(e_k, \hat{h})\}$
13:     **else**
14:         Let $(e^{\max}, h^{\max}) = \arg\max_{(e,h) \in \mathcal{C}_{\hat{y}}} h$
15:         **if** $\hat{h} < h^{\max}$ **then**
16:             Replace $(e^{\max}, h^{\max})$ with $(e_k, \hat{h})$
                                                    ▷ Phase 4: Similarity Computation
17:     **for** $j = 1$ **to** $|\mathcal{Y}_u|$ **do**
18:         **for all** $(k_{j,i}, h_{j,i}) \in \mathcal{C}_j$ **do**
19:             $a_{j,i}^{(d)} \leftarrow \exp\big[-\beta\big(1 - \cos(q^{(d)}, k_{j,i}^{(d)})\big)\big]$
20:     $o^{(d)} \leftarrow a^{(d)} Y$ for each descriptor $d$       ▷ Phase 5: LLM-Guided Adaptation
21:     Generate normalized weight $w \in \mathbb{R}^{P+Z+1}$ via LLM
22:     $s \leftarrow w \cdot \text{concat}(o^{(d)})$
23:     $\hat{y} \leftarrow \arg\max\big(\text{softmax}(\hat{\varphi} + \alpha_s s)\big)$

---

**Notation.**   We denote the original zero-shot logits produced by the frozen SZAR model as $\hat{\phi}$. The enhanced logits, augmented by Skeleton-Cache, are defined as $\varphi = \hat{\phi} + \alpha_s \mathbf{s}$ (see Eq. 12 in the main paper). The entropy of a probability distribution is denoted by $\mathcal{H}(\cdot)$. The cache for class $j$ is represented as $\mathcal{C}_j$, containing tuples of feature representations (cache keys $\mathbf{e}_k$), predicted labels

$(\mathbf{e}_y)$, and confidence scores $(\mathbf{e}_h)$. The descriptors include global $(\mathbf{g})$, spatial $(\mathbf{s}_p)$, and temporal $(\mathbf{t}_z)$ components, concatenated to form the cache key $\mathbf{e}_k$. The pseudocode outlines the Skeleton-Cache method, detailing the feature extraction, cache update, similarity computation, and LLM-guided fusion processes.

## B    Implementation Details

This section provides comprehensive details on the implementation of Skeleton-Cache, including efficiency analysis, the LLM prompt template for weight extraction, and our spatial and temporal partitioning strategies. These details ensure reproducibility and clarify the computational overhead and practical considerations of our method.

### B.1    Efficiency Analysis

As mentioned in Section 4.4, Table 5 presents the memory footprint and latency overhead of Skeleton-Cache across four zero-shot splits. The per-class memory consumption ranges from 120.0 KB to 128.0 KB, with the total memory requirement not exceeding 2.94 MB even for the largest split (NTU120 96/24). The computational latency varies from 0.40 seconds for the smallest split (NTU60 55/5) to 6.21 seconds for the largest (NTU120 96/24), reflecting the increased processing demands of cache operations as the number of unseen classes grows. These metrics confirm that Skeleton-Cache introduces minimal overhead compared to the base SZAR model, making it a lightweight solution suitable for real-time applications.

Table 5: Efficiency overhead of *Skeleton-Cache* on four zero-shot splits.

| Split | Unseen classes | Memory per class (KB) | total (MB) | Extra latency (s) |
|---|---|---|---|---|
| NTU60 55/5 | 5 | 128.0 | 0.64 | 0.40 |
| NTU60 48/12 | 12 | 120.0 | 1.44 | 1.98 |
| NTU120 110/10 | 10 | 128.0 | 1.28 | 2.27 |
| NTU120 96/24 | 24 | 122.7 | 2.94 | 6.21 |

### B.2    Prompt Template for LLM-Guided Weights

Skeleton-Cache Prompt (per action class)

You are an expert in human–action understanding. Given the action class `<ACTION>`, answer the following three questions **without adding commentary**.

1. **Spatial importance.** The human body is divided into four regions: [`Head, Torso, Arms, Legs`]. Provide a list of four non-negative numbers that sum to 1, corresponding to the relative importance of each region for recognising `<ACTION>`.
   Format: `"spatial":  [w_head, w_torso, w_arms, w_legs]`

2. **Temporal importance.** The action sequence is divided into three phases: [`Beginning, Middle, End`]. Provide a list of three non-negative numbers that sum to 1, indicating the relative importance of each phase.
   Format: `"temporal":  [w_begin, w_mid, w_end]`

3. **Global vs local preference.** Provide a single number $\gamma \in [0, 1]$ indicating how much the action should be recognised holistically ($\gamma \approx 1$) versus by local parts/phases ($\gamma \approx 0$).
   Format: `"gamma":  `$\gamma$

Return **one compact JSON object** with keys `"spatial"`, `"temporal"`, and `"gamma"`. Do not include any other keys, text, or explanations.

```
Example Response (action = "Waving")

{
  "spatial":  [0.05, 0.10, 0.70, 0.15],
  "temporal": [0.15, 0.60, 0.25],
  "gamma":    0.30
}
```

**LLM-guided weight extraction workflow.** To obtain the dataset-specific fusion weights, we employ a streamlined process leveraging LLM-based knowledge extraction. For each action class in the target dataset, we first construct a class-specific query by inserting the action name into our prompt template. These queries are then batch-processed through an LLM API (we use `gpt-4-turbo` with temperature $\tau = 0$), which returns structured JSON responses containing three key components: spatial importance weights $\mathbf{w}_{\mathrm{spa}}^{(c)} \in \mathbb{R}^P$ across body regions, temporal importance weights $\mathbf{w}_{\mathrm{tmp}}^{(c)} \in \mathbb{R}^Z$ across action phases, and a global-local preference parameter $\gamma^{(c)} \in [0, 1]$. We then process these raw outputs to construct a per-class weight vector $\widetilde{\mathbf{w}}^{(c)} = [\gamma^{(c)}, \ (1-\gamma^{(c)}) \cdot \mathbf{w}_{\mathrm{spa}}^{(c)\top}, \ (1-\gamma^{(c)}) \cdot \mathbf{w}_{\mathrm{tmp}}^{(c)\top}]$, followed by $\ell_1$-normalisation to obtain the final vector $\mathbf{w}^{(c)}$. The complete LLM-prior matrix $\mathbf{W} \in \mathbb{R}^{|\mathcal{C}| \times (P+Z+1)}$ is formed by stacking these normalized weight vectors for all classes and stored as a NumPy array for efficient access during test-time inference. This entire process is performed once per dataset without requiring any model training or fine-tuning, making it computationally efficient and broadly applicable across different skeleton-based action recognition tasks.

### B.3 Spatial and Temporal Partitioning

This subsection describes the partitioning strategy for spatial and temporal descriptors used in Skeleton-Cache for the NTU RGB+D and PKU-MMD datasets. The spatial partitioning divides the skeleton joints into semantically meaningful groups, while the temporal partitioning employs a uniform segmentation approach to capture action dynamics.

**Spatial Partitioning.** For both NTU RGB+D and PKU-MMD datasets, we partition the skeleton joints into four spatial groups corresponding to distinct body regions: head, torso, arms and feet. These groups are defined based on the joint indices provided in the datasets, ensuring that each group captures anatomically relevant features for action recognition. The specific joint indices for each spatial group in the NTU RGB+D dataset are as follows:

- **Head**: Joints {2, 3, 4, 8, 20}, covering the head and upper neck regions, with additional interpolation to include seven joints for robust feature extraction.
- **Torso**: Joints {0, 1, 4, 8, 12, 16, 20}, including the spine, shoulders, and central joints, with interpolation to include five additional joints for comprehensive torso representation.
- **Arms**: Joints {4, 5, 6, 7, 8, 9, 10, 11, 21, 22, 23, 24}, encompassing both hands and wrists to capture fine-grained hand movements critical for actions like "writing" or "waving."
- **Feet**: Joints {0, 12, 13, 14, 15, 16, 17, 18, 19}, covering the lower body, including hips, knees, and ankles, with interpolation to include three additional joints to enhance leg motion capture.

For the PKU-MMD dataset, the same semantic partitioning is applied, mapping the corresponding joint indices to these four regions while accounting for any dataset-specific differences in joint definitions. The spatial features $\mathbf{s}_p$ for each group $p \in \{1, 2, 3, 4\}$ are computed by averaging the feature representations $\mathbf{F}_{:,t,v}$ over the joints $v \in V_p$ and time steps $t \in [1, T]$, as described in Algorithm 1. This partitioning ensures that Skeleton-Cache captures localized motion cues relevant to specific body parts, enhancing the discriminative power of the descriptors.

**Temporal Partitioning.** The temporal partitioning strategy divides each skeleton sequence into three equal segments to capture the dynamics of the action across its progression: beginning, middle, and end phases. For a sequence with $T$ frames, the temporal segments are defined as follows:

- **Beginning**: Frames $t \in [1, \lfloor T/3 \rfloor]$.

- **Middle**: Frames $t \in [\lfloor T/3 \rfloor + 1, \lfloor 2T/3 \rfloor]$.
- **End**: Frames $t \in [\lfloor 2T/3 \rfloor + 1, T]$.

For each segment $z \in \{1, 2, 3\}$, the temporal features $\mathbf{t}_z$ are computed by averaging the feature representations $\mathbf{F}_{:,t,v}$ over the frames $t \in T_z$ and joints $v \in [1, V]$, as outlined in Algorithm 1. This uniform segmentation ensures that each phase of the action is equally represented, allowing Skeleton-Cache to model temporal evolution without requiring manual annotation of action phases. The approach is computationally efficient and generalizes well across datasets with varying sequence lengths, as demonstrated in our experiments on NTU RGB+D and PKU-MMD.

## C    More Quantitative Results

To further assess the robustness and generalizability of Skeleton-Cache, this section broadens the quantitative evaluation with additional zero-shot learning methods, comparisons against generic test-time adaptation baselines, and analysis of cache update strategies with adapted logits.

### C.1    ZSL Performance with Additional Methods

While the main paper evaluates Skeleton-Cache on several established ZSL methods, we extend the analysis here by experimenting with four additional ZSL methods on the NTU RGB+D 60 and NTU RGB+D 120 datasets. These experiments reinforce the trend observed in the main paper: more advanced ZSL methods exhibit greater performance improvements when augmented with Skeleton-Cache. This is attributed to Skeleton-Cache's reliance on high-confidence samples for populating the cache model, as advanced ZSL methods typically produce predictions with lower entropy compared to earlier methods, which often suffer from high entropy and class imbalance issues, such as misclassifying all samples of one class into another.

Table 6: Comparison of ZSL accuracy (%) on NTU RGB+D datasets. Our proposed Skeleton-Cache (**SC**) method shows greater improvements on more advanced ZSL methods due to their ability to generate more confident predictions for caching.

| | NTU RGB+D 60 | | NTU RGB+D 120 | |
|---|---|---|---|---|
| **Method** | **55/5 Split** | **48/12 Split** | **110/10 Split** | **96/24 Split** |
| ReViSE [9] | 53.91 | 17.49 | 55.04 | 32.38 |
| JPoSE [28] | 64.82 | 28.75 | 51.93 | 32.44 |
| CADA-VAE [19] | 76.84 | 28.96 | 59.53 | 35.77 |
| SynSE [8] | 75.81 | 33.30 | 62.69 | 38.70 |
| SMIE [32] | 77.98 | 40.18 | 65.74 | 45.30 |
| PURLS [33] | 79.22 | 40.99 | 71.95 | 52.01 |
| SA-DVAE [15] | 82.37 | 41.38 | 68.77 | 46.12 |
| STAR [3] | 81.40 | 45.10 | 63.30 | 44.30 |
| ReViSE+**SC** | **54.70**↑0.79 | **18.05**↑0.56 | **55.83**↑0.79 | **33.01**↑0.63 |
| JPoSE+**SC** | **66.31**↑1.49 | **29.84**↑1.09 | **53.44**↑1.51 | **33.56**↑1.12 |
| CADA-VAE+**SC** | **78.94**↑2.10 | **30.49**↑1.53 | **61.53**↑2.00 | **37.30**↑1.53 |
| SynSE+**SC** | **79.73**↑3.92 | **38.82**↑5.52 | **68.47**↑5.78 | **44.10**↑5.40 |
| SMIE+**SC** | **82.63**↑4.65 | **44.17**↑3.99 | **72.98**↑7.24 | **50.44**↑5.14 |
| PURLS+**SC** | **85.46**↑6.24 | **45.22**↑4.23 | **77.60**↑5.65 | **56.83**↑4.82 |
| STAR+**SC** | **88.82**↑7.42 | **52.03**↑6.93 | **69.54**↑6.24 | **50.02**↑5.72 |
| SA-DVAE+**SC** | **89.41**↑7.04 | **47.83**↑6.45 | **74.29**↑5.52 | **53.14**↑7.02 |

Table 6 compares the Top-1 accuracy of eight ZSL methods, including four additional methods (ReViSE, JPoSE, CADA-VAE, and STAR) not evaluated in the main paper, on four zero-shot splits of NTU RGB+D 60 and NTU RGB+D 120. The results show that Skeleton-Cache (**SC**) consistently improves performance across all methods, with larger gains observed for more advanced methods like SA-DVAE and PURLS. For instance, SA-DVAE+SC achieves a 7.04% improvement on the NTU60 55/5 split, compared to only a 2.79% improvement for ReViSE+SC. This trend is consistent across splits, as advanced methods generate more confident predictions, enabling Skeleton-Cache to effectively cache representative exemplars. In contrast, earlier methods like ReViSE and JPoSE

often produce high-entropy predictions or collapse to predicting a single class, limiting the cache's effectiveness.

## C.2 Comparison with Generic TTA Baselines

Table 7 contrasts **Skeleton-Cache** with mainstream test-time adaptation (TTA) methods—gradient-based prompt-tuning (TPT [24], DiffTPT [6]), training-free prototype or attention modulation (AdaNPC [31], CALIP [7]) and dual-cache retrieval (TDA [11])—using a shared frozen *PURLS* backbone. By leveraging skeleton-specific priors, our method arranges each sequence into eight structured descriptors (global, four spatial parts, three temporal phases) and fuses their similarities under class-wise weights produced by a single LLM prompt. This dedicated design preserves topology, mitigates viewpoint drift, and consistently delivers the best accuracy on every NTU-RGBD split without back-propagation or synthetic augmentations.

In contrast, existing baselines exhibit limitations when applied to skeleton data. Gradient-based prompt tuning (TPT, DiffTPT) requires iterative optimisation that easily over-fits the small and highly imbalanced target streams. Prototype/attention schemes (AdaNPC, CALIP) compress each clip into a single holistic vector, discarding fine-grained joint dynamics and thus confusing actions that differ only in local motion. TDA partially addresses this with a dual cache, yet still operates at a single global scale and ignores temporal phase diversity. None of these methods explicitly model skeleton topology or spatial-temporal structure, which explains their inferior performance compared with the skeleton-aware retrieval strategy proposed here.

Table 7: Comparison with other TTA methods on NTU-RGBD datasets.

| Method | NTU-RGBD 60 | | NTU-RGBD 120 | |
| --- | --- | --- | --- | --- |
| | 55/5 | 48/12 | 110/10 | 96/24 |
| TPT [24] | 40.11 | 27.98 | 22.10 | 17.10 |
| DiffTPT [6] | 42.27 | 28.34 | 23.43 | 20.01 |
| AdaNPC[31] | 81.77 | 42.52 | 72.50 | 53.16 |
| CALIP [7] | 80.02 | 45.10 | 72.27 | 51.84 |
| TDA[11] | 82.84 | 43.95 | 74.92 | 54.60 |
| **Ours** | **85.46** | **45.22** | **77.60** | **56.83** |

## C.3 Cache Update with Adapted Logits

To explore the impact of cache update strategies, we conducted an experiment where Skeleton-Cache is updated using the final adapted logits (post-fusion, as defined in Equation 12) instead of the zero-shot logits from the PURLS model. This approach leverages the enhanced predictions after combining cache-based similarity logits with the original model logits, potentially incorporating more refined confidence estimates. Table 8 compares the Top-1 accuracy of this adapted-logits strategy

Table 8: Top-1 accuracy (%) of Skeleton-Cache with cache updates using adapted logits versus zero-shot logits on NTU RGB+D datasets.

| Split | SC (Zero-Shot Logits) | SC (Adapted Logits) | Difference |
| --- | --- | --- | --- |
| NTU60 55/5 | 85.46 | 85.77 | +0.31 |
| NTU60 48/12 | 45.22 | 44.66 | -0.56 |
| NTU120 110/10 | 77.60 | 77.84 | +0.24 |
| NTU120 96/24 | 56.83 | 56.90 | +0.07 |

against the standard Skeleton-Cache (SC) approach, which uses zero-shot logits for cache updates, across four zero-shot splits of NTU RGB+D 60 and 120. The results show minimal differences, with adapted-logits accuracies of 85.77%, 44.96%, 77.84%, and 56.90% compared to 85.46%, 45.22%, 77.60%, and 56.83% for the standard approach. Notably, a slight performance decline is observed on the NTU60 48/12 split (44.96% vs. 45.22%). This may stem from reduced overall entropy in the adapted logits, as the fusion process (Section 3.2) enhances prediction confidence, potentially allowing

suboptimal samples to enter the cache despite the high-confidence selection criterion. These samples may not represent the class as effectively, impacting retrieval accuracy. The limited performance variation across other splits suggests that the high-confidence filtering mechanism (Section 3.2) is robust, mitigating significant deviations when using adapted logits.

# D    Analytical Visualizations

This section presents visualizations to illustrate the impact of Skeleton-Cache on prediction quality, focusing on confusion matrices, entropy shifts, and LLM-derived weight distributions. These analyses provide insights into how the method improves class discrimination and confidence in predictions.

## D.1    LLM Weight Heatmap Analysis

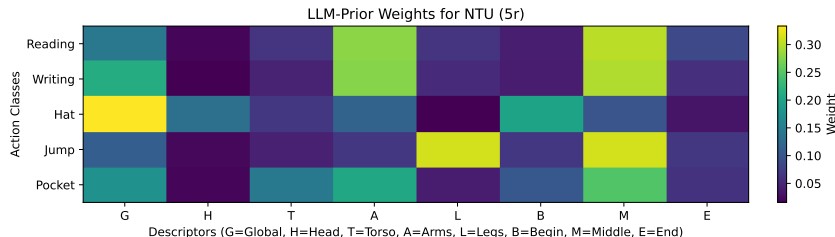

Figure 3: Heat-map visualisation of GPT-4–derived weights $\mathbf{w}^{(c)}$. Columns correspond to the eight descriptors; rows correspond to unseen classes of NTU 55/5 split.

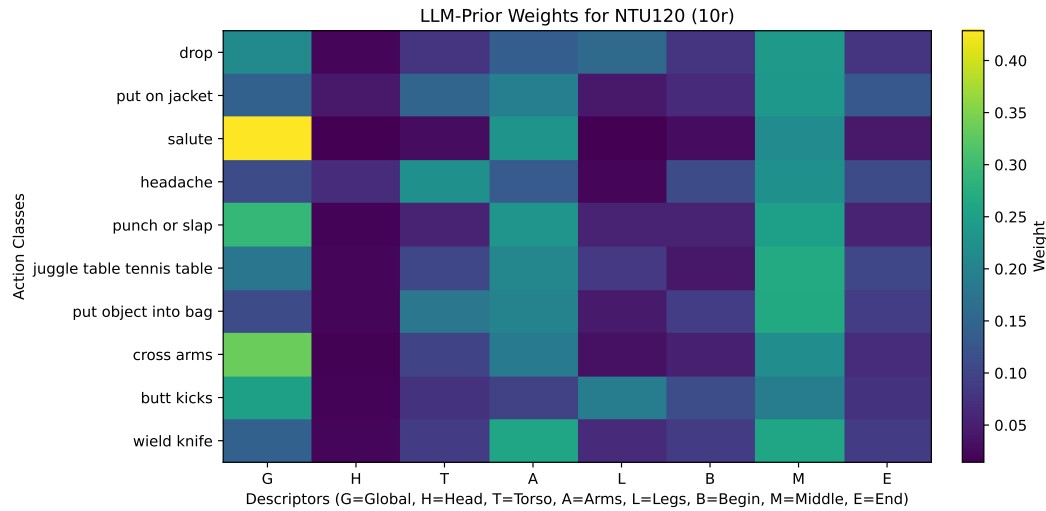

Figure 4: Heat-map visualisation of GPT-4–derived weights $\mathbf{w}^{(c)}$. Columns correspond to the eight descriptors; rows correspond to unseen classes of NTU 110/10 split.

Figures 3 and 4 display heatmaps of the GPT-4-derived weights $\mathbf{w}^{(c)}$ for the NTU 55/5 and 110/10 splits, respectively. These weights align closely with human intuition about actions. For example, in the NTU 55/5 split, *reading* and *writing* emphasize the *arms* descriptor, while *put on hat/cap* assigns significant weight (approximately 40%) to the *head*. Similarly, *jump up* focuses on *legs* and the *middle* temporal segment, reflecting the action's peak motion phase. This alignment with commonsense semantics enhances inter-class discriminability by assigning distinct weight patterns to different actions, enabling the model to differentiate between similar motion patterns effectively.

## D.2  Confusion Matrix Analysis

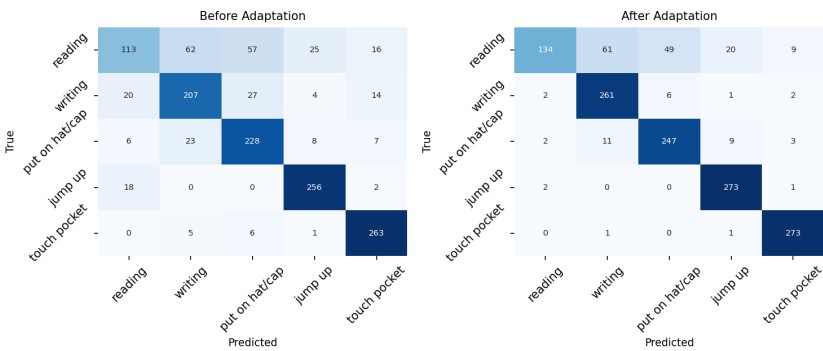

Figure 5: Comparison of confusion matrices on NTU 55/5 split. Adaptation with Skeleton-Cache sharpens the diagonal and reduces off-diagonal confusion, indicating improved class discrimination.

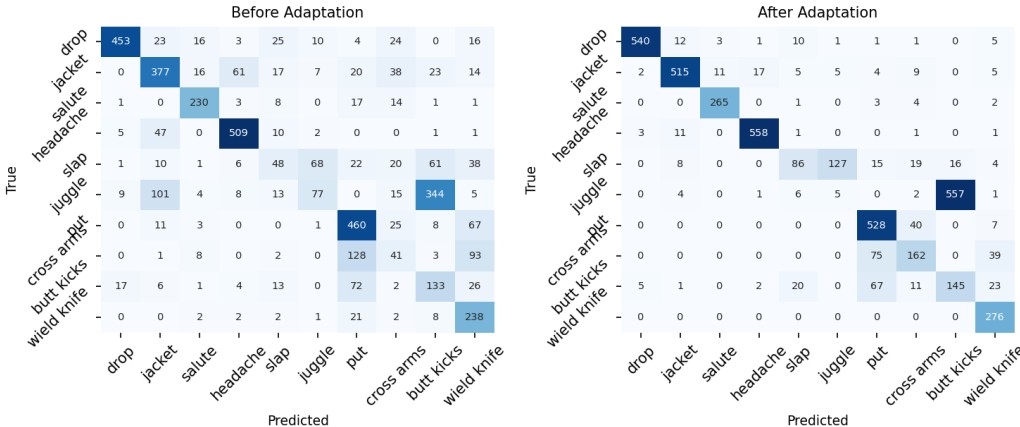

Figure 6: Comparison of confusion matrices on NTU 110/10 split. Adaptation with Skeleton-Cache sharpens the diagonal and reduces off-diagonal confusion, indicating improved class discrimination.

Figures 5 and 6 compare confusion matrices before and after adaptation with Skeleton-Cache for the NTU 55/5 and 110/10 splits. Before adaptation, the model struggles to distinguish between similar actions, such as *reading* and *writing*, due to their overlapping upper-body poses, resulting in significant off-diagonal confusion. Post-adaptation, the diagonal sharpens, and off-diagonal elements lighten, indicating that Skeleton-Cache enables differentiation of previously indistinguishable classes. However, this improvement introduces a trade-off: the model occasionally exhibits overconfidence, leading to misclassifications in some cases, as evidenced by residual off-diagonal elements.

## D.3  Top-5 Prediction Changes

Figure 7 visualizes the confidence score shifts in the top-5 predictions for 10 unseen classes in the NTU 110/10 split. Skeleton-Cache significantly enhances prediction confidence, particularly for challenging zero-shot tasks. For instance, actions like *salute* and *cross arms* show marked increases in confidence for the correct class, reducing uncertainty and refining the prediction distribution. This boost in confidence underscores Skeleton-Cache's ability to leverage structured descriptors and cache-based retrieval to improve recognition accuracy for unseen actions.

### D.4  Per-Class Accuracy Changes Before and After Adaptation

To elucidate the impact of Skeleton-Cache (SC) on per-class performance, we visualize the Top-1 accuracy changes for the NTU RGB+D 55/5 and 110/10 splits before and after test-time adaptation with PURLS+SC (Section 3.2). Figure 8 compares the base PURLS accuracies against the adapted accuracies, highlighting the contribution of the cache-based retrieval and LLM-guided fusion.

For the NTU 55/5 split (Figure 8a), Skeleton-Cache consistently improves accuracy across all classes, with notable gains in *writing* (76.10% to 95.96%) and *reading* (41.39% to 49.08%). These improvements stem from the structured descriptors capturing fine-grained motion patterns (e.g., hand movements) and the LLM-guided weighting emphasizing relevant body parts (Section 3.2). For the NTU 110/10 split (Figure 8b), most classes show substantial gains, such as *cross arms* (14.86% to 58.70%) and *slap* (17.45% to 31.27%), benefiting from the cache's ability to retrieve discriminative local patterns. However, *juggle* exhibits a significant drop (13.37% to 0.87%), likely due to its complex, multi-limb coordination, which challenges the cache's retrieval when high-confidence exemplars are scarce. Overall, Skeleton-Cache enhances generalization for most classes, with occasional limitations for actions requiring intricate motion patterns.

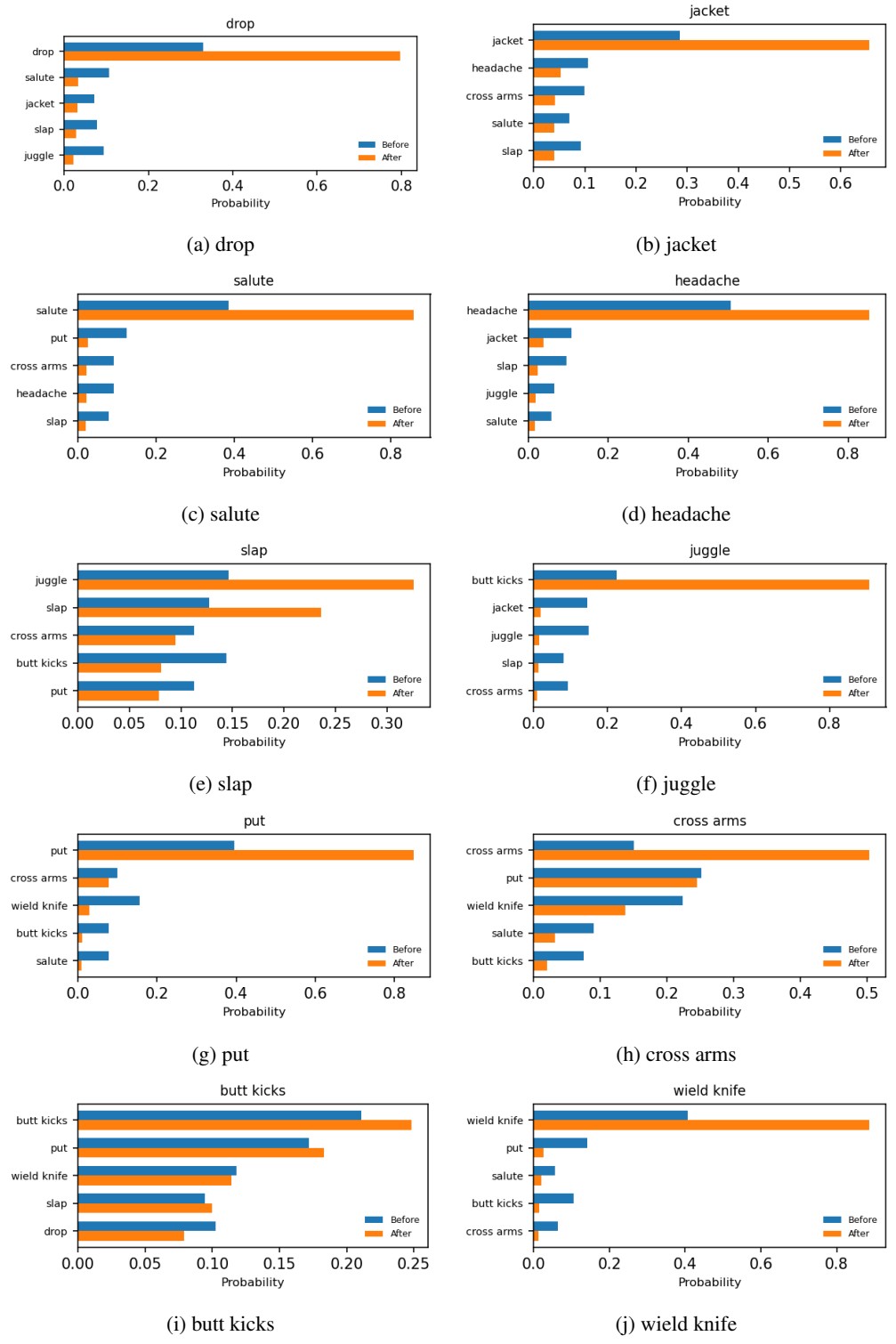

Figure 7: Visualization of Top-5 prediction changes before and after applying the Skeleton-Cache for 10 different unseen action classes. Each image displays the confidence score distribution shift when using our caching mechanism, demonstrating how the prediction accuracy improves for challenging zero-shot action recognition tasks.

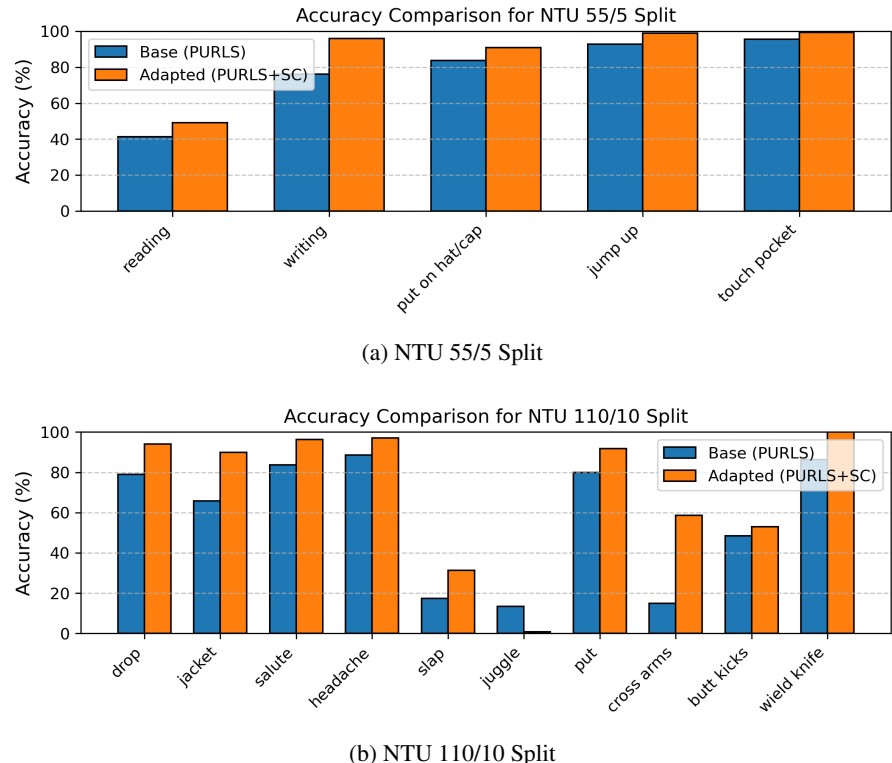

(a) NTU 55/5 Split

(b) NTU 110/10 Split

Figure 8: Per-class accuracy comparison between base PURLS and adapted PURLS+SC for NTU 55/5 and 110/10 splits. Blue bars represent base accuracies, and orange bars represent adapted accuracies.

