# OpenReview forum: "Boosting Skeleton-based Zero-Shot Action Recognition with Training-Free Test-Time Adaptation"
_NeurIPS.cc/2025/Conference — NeurIPS 2025 poster_

### Official Review · Reviewer_LMYG · 2025-06-21

**Clarity:** 3
**Significance:** 2
**Originality:** 2
**Rating:** 4
**Confidence:** 4

**Summary:**

This paper introduces Skeleton-Cache, a training-free test-time adaptation framework for skeleton-based zero-shot action recognition. By dynamically constructing and updating a non-parametric cache during testing, the model adapts to unseen actions without additional training. Experimental results demonstrate that Skeleton-Cache significantly improves recognition accuracy across multiple benchmark datasets, showcasing its effectiveness and efficiency in zero-shot learning.

**Questions:**

Would the sequence of the samples influence the test results.

**Ethical Concerns:**

["NO or VERY MINOR ethics concerns only"]

**Final Justification:**

The authors addressed most of my concerns, and I will keep my recommendation.

**Quality:**

3

**Strengths And Weaknesses:**

Strengths:
1. Innovative Training-Free Test-Time Adaptation. The Skeleton-Cache module operates without additional training, dynamically updating a non-parametric cache at inference time to improve generalization to unseen actions. This design is both practical and efficient, enabling real-time deployment and serving as a plug-and-play enhancement for various skeleton-based action recognition models.
2. LLM-Guided Semantic Fusion. By leveraging a large language model to assign class-specific importance weights to different body regions or temporal segments, the method enhances focus on the most relevant features for unseen actions. This novel use of semantic reasoning to guide descriptor fusion helps the model better emphasize critical skeleton regions and motion phases in zero-shot scenarios, leading to improved recognition performance.

Weaknesses
1. Dependence on Pre-trained Feature Extractors. The approach relies heavily on the quality of pre-trained skeleton encoders. If the underlying backbone struggles with noisy, occluded, or low-quality pose estimates, Skeleton-Cache’s adaptation will be limited by those upstream errors.
2. Cache Memory Overhead. Although non-parametric, the dynamic cache can grow large during long inference sessions or on large-scale datasets. Without careful management, this may introduce nontrivial memory and runtime overhead in resource-constrained deployments.
3. Reproducibility Concerns. The paper states that code release will follow acceptance, but no pre-release implementation is available. This delays independent verification and hinders rapid adoption by the community.

---

> ### Author Rebuttal · Authors · 2025-07-30
>
> We sincerely thank you for your valuable time and constructive feedback. We are encouraged that you recognized our framework's innovation and applicability. We address your concerns below.
>
> ### W1: Dependence on Pre-trained Feature Extractors Quality Limitations
>
> We appreciate this thoughtful observation regarding the fundamental dependence on backbone quality—a challenge that extends beyond our method to the broader skeleton-based action recognition field. Indeed, when skeleton pose estimation is severely compromised, all downstream methods face inherent limitations. However, we would like to highlight several design aspects of Skeleton-Cache that provide inherent robustness to these challenges: our structured representation naturally decomposes skeleton sequences into multi-scale descriptors (global, spatial, temporal), providing redundancy when certain body parts are occluded or noisy; the LLM-guided fusion adaptively weights descriptor contributions based on semantic priors, naturally downweighting unreliable features from compromised body regions; and our entropy-based cache selection acts as a confidence filter, preferentially storing high-quality samples while discarding uncertain predictions. These mechanisms collectively help maximize feature utility even when the backbone occasionally produces suboptimal representations.
>
> Our experimental results demonstrate that Skeleton-Cache provides consistent improvements across diverse backbone architectures with varying baseline performance levels. As shown in Tables 1, 2, and 6, we achieve substantial gains on both established methods like SynSE (+3.92% on NTU60 55/5) and state-of-the-art approaches like SA-DVAE (+7.04% on the same split). These improvements hold across different data splits and datasets, suggesting that our framework effectively enhances feature utilization within each backbone's representational capacity. We acknowledge that addressing severe pose estimation failures remains an open challenge for the entire skeleton-based recognition community, but our contribution lies in demonstrating how structured representation and adaptive fusion can maximize the utility of available features, regardless of the underlying backbone's limitations.
>
> ### W2: Cache Memory Overhead in Resource-Constrained Deployments
> This is a very practical concern. We have carefully analyzed the memory footprint and the results confirm that the overhead is minimal. For clarity, we present the full data below:
>
> |Split|Unseen Classes|Memory per Class (KB)|Total Memory (MB)|Sample Memory(MB)|Sample Number|
> |---|---|---|---|---|---|
> |NTU60 55/5|5|128.0|0.64|105.1|1368|
> |NTU60 48/12|12|120.0|1.44|252.7|3291|
> |NTU120 110/10|10|128.0|1.28|326.3|4249|
> |NTU120 96/24|24|122.7|2.94|759.7|9892|
>
>
> As shown, the total memory required peaks at only 2.94 MB for the largest split with 9892 samples (759.7 MB). Compared to the sample memory, the cache memory overhead is minimal. This analysis demonstrates that Skeleton-Cache is a lightweight solution suitable for real-world deployment where resources may be constrained.
>
> ### W3: Reproducibility Concerns - No Pre-release Implementation Available
>
> We fully agree that reproducibility is paramount. We apologize that we cannot provide the link during the rebuttal stage due to the latest NeurIPS 2025 policy restrictions. However, we are fully committed to releasing our complete source code, pre-trained features, and weight files upon acceptance. To facilitate replication in the meantime, we have provided extensive implementation details in the appendix. This includes:
>
> (1) Detailed pseudocode of the entire pipeline in Algorithm 1.
> (2) The exact LLM prompt template used for weight generation in Appendix B.2.
> (3) Precise definitions of our spatial and temporal partitions for feature extraction in Appendix B.3.
>
> We believe these resources provide a clear blueprint for reproducing our results, and we will release our code as soon as possible.
>
> ### Q1: Impact of Sample Sequence on Test Results
>
> Thank you for this insightful question. Our system is designed for high robustness against sample order. As detailed in Section 3.2, "Cache Construction and Update", the cache is populated based on prediction confidence, where high-confidence samples are added and low-confidence ones are replaced. This ensures the cache quickly stabilizes with representative class exemplars, making its final state largely insensitive to the initial sample sequence. To empirically validate this, we evaluated the PURLS backbone on four splits, each with five different random shuffles of the test set. The results show a negligible standard deviation, confirming that performance is highly robust and not dependent on sample order.
>
> |Run (Random Seed)|NTU60 55/5|NTU60 48/12|NTU120 110/10|NTU120 96/24|
> |---|---|---|---|---|
> |1|85.67|45.32|77.01|56.64|
> |2|85.18|45.62|77.33|56.86|
> |3|84.96|45.12|77.33|56.73|
> |4|85.32|44.94|77.73|57.02|
> |5|85.69|45.46|77.41|56.43|
> |**Mean ± Std. Dev.**|**85.36 ± 0.32**|**45.29 ± 0.27**|**77.36 ± 0.26**|**56.74 ± 0.22**|
>
> Thank you again for your valuable feedback, which we believe has helped strengthen our paper.

---

### Official Review · Reviewer_27pj · 2025-06-24

**Clarity:** 4
**Significance:** 3
**Originality:** 3
**Rating:** 5
**Confidence:** 3

**Summary:**

This paper proposes Skeleton-Cache, a novel training-free test-time adaptation framework designed for skeleton-based zero-shot action recognition. Instead of relying on fixed representations, Skeleton-Cache reformulates inference as a retrieval task over a non-parametric feature cache. Each incoming skeleton sequence is encoded into a structured set of descriptors which are stored in class-specific cache entries. During testing, the model retrieves similar descriptors from the cache to refine its predictions on-the-fly. Importantly, the framework leverages large language models to assign semantic importance weights to each descriptor type. This LLM-guided weighting adaptively fuses the descriptor-level predictions, highlighting the most informative body regions or motion phases for each action. Skeleton-Cache integrates seamlessly with existing zero-shot skeleton action recognition models as a plug-and-play module and requires no additional training or gradient updates during deployment. Experiments on standard benchmarks show that adding Skeleton-Cache consistently boosts accuracy on unseen action classes under both conventional zero-shot and generalized zero-shot settings. The proposed method thus addresses the distribution shift between seen and unseen actions by dynamically adapting to test data without retraining, and achieves new state-of-the-art results in SZAR.

**Questions:**

None

**Ethical Concerns:**

["NO or VERY MINOR ethics concerns only"]

**Final Justification:**

The authors have addressed my concerns, and I will keep my recommendation.

**Quality:**

3

**Strengths And Weaknesses:**

Strength
1. The paper is well-written and easy to follow.
2. The idea of applying test-time adaptation in zero-shot skeleton action recognition is novel. The paper identifies a clear gap, as existing SZAR methods cannot adapt at inference. The authors introduce Skeleton-Cache as the first training-free TTA module for the task, reformulating skeleton action recognition as a cache-based retrieval problem, combined with a structured, multi-scale representation of skeleton data.
3. The experimental section is thorough and convincing. Results on three benchmarks (NTU-60, NTU-120, PKU-MMD) and multiple data splits show consistent improvements across the board. For instance, aattaching Skeleton-Cache to the SA-DVAE model boosts its zero-shot accuracy on NTU-60/120 by ~6% in top-1 accuracy.
4. Skeleton-Cache is a training-free, plug-and-play module that can work with different backbone models, which is a practical strength. It does not alter the learned model parameters and introduces only a tiny overhead.

weakness
1. The paper could offer a deeper analysis of when and why the method might fail or underperform. For instance, one potential concern is how Skeleton-Cache handles misclassifications during the adaptation process. Since the cache is updated using the model’s predicted labels (which could occasionally be wrong for unseen classes), there is a risk that an early misprediction could insert an incorrect prototype into the cache, possibly confusing subsequent retrievals.
2. The absence of code implementation may undermine the reproducibility of the proposed method.

---

> ### Author Rebuttal · Authors · 2025-07-31
>
> We sincerely thank you for your positive and detailed review. We are greatly encouraged by your recognition of our work's novelty, clarity, and experimental thoroughness. We appreciate the insightful points you raised and offer the following clarifications.
>
> ### W1: Deeper Analysis of Failure Cases and Misclassification Handling
>
> We sincerely thank the reviewer for this insightful and important observation. As noted by the reviewer, our Skeleton-Cache mechanism relies on predicted labels to update the prototype cache. If misclassifications occur early, there is a risk that incorrect prototypes are inserted into the cache, potentially misleading future predictions. We have observed this behavior manifest in two primary failure modes:
>
> **Failure Mode 1: Confusion Between Semantically Similar Actions**
> In some scenarios, actions like reading and writing share overlapping joint dynamics and can be easily confused by the backbone model, as shown in the following confusion matrix. The early mispredictions can propagate through the cache.
>
> To address this, in our skeleton-cache (SC), we first employ entropy-based confidence filtering to ensure that only high-confidence predictions are considered for caching, excluding uncertain predictions from the cache. Second, we implement a dynamic replacement policy that continuously monitors cache quality and replaces the least confident entries when the cache reaches capacity. These mechanisms work in tandem to maintain high-quality prototypes: the entropy filtering prevents low-confidence mispredictions from entering, while the replacement policy ensures that any early confident but incorrect samples are gradually displaced by higher-quality examples over time, directly mitigating the risk of performance degradation from initial mispredictions.
>
> While we observe some improvements in the diagonal elements (reading: 113→134, writing: 207→261), the cross-confusion remains substantial. Specifically, 61 reading samples (22.3%) are still misclassified as writing after adaptation, demonstrating the limited effectiveness when incorrect prototypes contaminate the cache early in the process.
>
> **Before Adaptation**
>
> |True\Predicted|reading|writing|put_on_hat/cap|jump_up|touch_pocket|
> |---|---|---|---|---|---|
> |**reading**|113|62|57|25|16|
> |**writing**|20|207|27|4|14|
> |**put_on_hat/cap**|6|23|228|8|7|
> |**jump_up**|18|0|0|256|2|
> |**touch_pocket**|0|5|6|1|263|
>
> **After Adaptation**
>
> |True\Predicted|reading|writing|put_on_hat/cap|jump_up|touch_pocket|
> |---|---|---|---|---|---|
> |**reading**|134|61|49|20|9|
> |**writing**|2|261|6|1|2|
> |**put_on_hat/cap**|2|11|247|9|3|
> |**touch_pocket**|0|1|0|1|273|
>
> **Failure Mode 2: Amplified Errors from Occlusion and Noise**
> Actions involving complex hand interactions, self-occlusions, rapid movements, and subtle gestures produce inherently noisy features that can be misclassified with high confidence due to model overconfidence. When these incorrect but confident predictions are cached, they actively degrade performance by providing misleading retrieval candidates. In the NTU120 96/24 split, such challenging actions show limited improvements:
>
> While our method provides improvements, these actions remain significantly below the overall average accuracy of 56.83%, indicating the substantial gap that persists. Particularly for actions with very poor initial recognition (e.g., thumb up), the improvements are minimal because our mechanism fundamentally depends on the backbone model's representational quality.
>
> |Action|Before Adaptation|After Adaptation|
> |---|---|---|
> |pick up|26.60%|40.86%|
> |take off hat or cap|15.38%|35.54%|
> |rock paper scissors|15.45%|22.46%|
> |thumb up|2.04%|2.77%|
>
> ### W2: Code Availability
>
> We completely agree that reproducibility is important. As stated at the end of our abstract (line 13), we have always planned to release our code. We reaffirm our commitment to make our work fully reproducible. We apologize that we cannot provide the link during the rebuttal stage due to the latest NeurIPS 2025 policy restrictions. We will release the complete source code on a public GitHub repository. This will include:
>
> (1) The full implementation of the Skeleton-Cache module.
> (2) Pre-generated LLM-guided prior files for all datasets.
> (3) Scripts to reproduce the main results reported in our paper.
>
> We will add a clear link to this repository in the final version of the paper.
>
> Thank you once again for your constructive feedback and your confidence in our work. We hope these clarifications address your concerns and demonstrate our commitment to thorough evaluation and reproducibility.

---

> > ### Comment · Reviewer_27pj · 2025-08-08
> >
> > The authors have addressed my concerns, and I will maintain my recommendation.

---

### Official Review · Reviewer_qjj5 · 2025-06-28

**Clarity:** 3
**Significance:** 3
**Originality:** 3
**Rating:** 4
**Confidence:** 4

**Summary:**

The paper introduces Skeleton-Cache, a novel framework designed to enhance skeleton-based zero-shot action recognition (SZAR). The key innovation is a training-free test-time adaptation (TF-TTA) module that dynamically adapts to unseen actions during inference without requiring additional training or access to training data. Skeleton-Cache uses a non-parametric cache to store structured skeleton representations, combining global and fine-grained local descriptors. The framework leverages large language models (LLMs) to assign class-specific importance weights, guiding the fusion of descriptor-wise predictions.

**Questions:**

1. Please clarify the retrieval process details. How is each component used for retrieval?
2. How is the weighting implemented? Is it class-specific or not?
3. What are the exact weight values? Did you tweak them a little bit after getting them from LLMs?
4. Please add a comparison of inference latency with other TF-TTA methods.

**Ethical Concerns:**

["NO or VERY MINOR ethics concerns only"]

**Final Justification:**

I thank the authors for providing detailed responses. My concerns are solved. I would like to keep my positive rating.

**Limitations:**

The authors could discuss more societal impact. As human data is sensitive, the authors should add more discussions about ethics, etc.

**Paper Formatting Concerns:**

Line 333 has a sudden change of line to Line 334 without completion.
Fig. 1 of $e_\hat{k}$ is a bit confusing; it would be better to be split.

**Quality:**

3

**Strengths And Weaknesses:**

Strengths:
1. Skeleton-Cache (SC) introduces a novel training-free test-time adaptation framework, which is the first TF-TTA method used in the field of SZAR. In addition, the authors considers local spatial, local temporal features, and a global feature for cache keys.
2. The method is computationally efficient, avoiding gradient-based updates and relying on a lightweight retrieval process.
3. Integration with spatial and temporal-specific weights from LLMs: The use of LLMs to derive spatial/temporal-specific importance weights is a unique approach that improves the accuracy of action recognition.
4. The paper is well written.

Weakness:
1. Some details of cache retrieval are not clear. From Fig. 1 and Eq. (4), I thought the vector $e_\hat{k}$ is used for retrieval. However, Eq.(6) suddenly introduced $k_{j,i}^d$. It took me several readings to understand $k_{i,j}^d$ might be one of $g$, $s$, $t$ used in Eq.(4).
2. The details of the weighting are not clear. In line 234, it seems the weighting is for each class. However, Eq.(9) seems the weighting is the same for all the classes.
3. The use of LLM may unnecessarily be overclaimed. These weights are essentially a set(s) of predefined weights. LLMs may hallucinate and may lead to worse performance.
3. The authors should report a comparison of inference latency.

---

> ### Author Rebuttal · Authors · 2025-07-30
>
> We sincerely thank you for your thoughtful and constructive review. Your detailed feedback is immensely helpful, and we appreciate the opportunity to clarify these points and improve our manuscript.
>
> ### W1 & Q1: Cache Retrieval Process Details Not Clear ($e_k$ vector vs $q^{(d)}$ notation)
>
> We sincerely apologize for the lack of clarity in our notation regarding the cache retrieval process. To clarify, the final cache key for a single sequence is a structured matrix $\mathbf{e}_{\hat{k}} \in \mathbb{R}^{(P+Z+1) \times N}$, which is formed by concatenating all individual descriptor vectors as you correctly noted:
>
> $\mathbf{e}_{\hat{k}} = \mathrm{concat}(g, s_1, ..., s_P, t_1, ..., t_Z)$
>
> To simplify notation, we use $q^{(d)}$ and $k_{j,i}^{(d)}$ to denote the $d$-th descriptor (row) of the query matrix and cache key matrix, respectively, where $j \in \{1, 2, ..., |Y_u|\}$ denotes the class index and $i \in \{1, 2, ..., K\}$ denotes the entry index within class $j$'s cache block.
>
> The retrieval process is performed in a strictly descriptor-wise manner. Given a test sample, we first extract its descriptors to form the query matrix. Then, for each descriptor type $d$, we compute the affinity between the query descriptor $q^{(d)}$ and all corresponding cached descriptors $k_{j,i}^{(d)}$ **across all entries in the cache (i.e., for all $j$ classes and all $i$ entries within each class, totaling up to $|Y_u| \times K$ comparisons)**. These affinity scores are used to weight the cached class labels, producing descriptor-wise predictions that are subsequently fused. In the revised manuscript, we will restructure **Section 3.2** to first define the full cache key as a matrix of descriptors, and then explicitly explain that the query and cache descriptors are simply the corresponding rows of their respective matrices, ensuring a logical flow from **Eq. (4)** to **Eq. (6)**.
>
> ### W2 & Q2: Weighting Implementation Details - Class-Specific or Same for All Classes?
>
> The weights derived from the LLM are indeed **class-specific**. As stated in **line 234**, "For each action category $c$ in $\mathcal{Y}_u$, we issue a single prompt to the LLM...". This procedure yields a unique set of raw importance scores for each individual class, which are then formulated into a class-specific weight vector $w^{(c)}$. In Eq.9, we did not add class notion for simplicity. We will revise **Eq. (9)** to eliminate ambiguity:
>
> $$w^{(c)} = \left[ \gamma^{(c)}, (1-\gamma^{(c)}) \cdot w_{\text{spa}}^{(c,1)}, \dots, (1-\gamma^{(c)}) \cdot w_{\text{spa}}^{(c,P)}, (1-\gamma^{(c)}) \cdot w_{\text{tmp}}^{(c,1)}, \dots, (1-\gamma^{(c)}) \cdot w_{\text{tmp}}^{(c,Z)} \right]$$
>
> The superscript $(c)$ on all components explicitly denotes their dependence on the action class $c$. This ensures, for example, that the action "writing" will have a weight vector $w^{(\text{writing})}$ that prioritizes the `arms` descriptor, while the action "jump up" will have a different vector $w^{(\text{jump up})}$ that emphasizes the `legs` and `middle` temporal phase descriptors. To further clarify, we will add a declarative sentence immediately following the revised **Eq. (9)** to re-emphasize that the weight vector $w^{(c)}$ is uniquely generated for each unseen class.
>
> ### W3 & Q3: LLM Role Overclaimed and Weight Values/Tweaking Details
>
> We believe it is critically important to avoid overclaiming the role of the LLM, and we thank you for raising this valid point. In our work, the LLM serves as a source of structured knowledge, generating class-specific semantic priors in a completely training-free manner within our test-time adaptation framework. The LLM's key contribution is providing nuanced, class-specific guidance that is otherwise absent, especially valuable in zero-shot settings. To maintain our training-free approach and ensure transparency, we obtain weight values for spatial importance, temporal importance, and global-vs-local preference directly from the LLM using the structured prompt below, without any manual adjustment or tuning. Notably, even without tuning these weights, our method already achieves significant improvements over various baselines, demonstrating the effectiveness of LLM-generated priors. We deliberately chose not to fine-tune these weights to preserve the training-free nature of our approach, though future work could explore weight optimization for potentially better performance.
>
> We acknowledge that LLMs can exhibit biases and hallucinations; however, the quantitative and qualitative results in our appendix show that these weights align well with human intuition about action recognition, empirically validated by the superior performance of LLM weights over uniform averaging. This LLM-guided weighting makes our adaptation more semantically aware, opening broader implications for domains like medical imaging or scientific discovery, where feature-based and attribute-based learning is limited by expensive expert annotation. Our method offers a pathway to inject domain knowledge without manual labeling. We will revise our discussion to frame the LLM's contribution more precisely in this context.
>
> **Prompt (per action class)**
>
> ```
> You are an expert in human-action understanding. Given the action class <ACTION>, answer the following three questions without adding commentary.
>
> 1. Spatial importance. The human body is divided into four regions: [Head, Torso, Arms, Legs]. Provide a list of four non-negative numbers that sum to 1, corresponding to the relative importance of each region for recognising <ACTION>.
> Format: "spatial": [w_head, w_torso, w_arms, w_legs]
>
> 2. Temporal importance. The action sequence is divided into three phases: [Beginning, Middle, End]. Provide a list of three non-negative numbers that sum to 1, indicating the relative importance of each phase.
> Format: "temporal": [w_begin, w_mid, w_end]
>
> 3. Global vs local preference. Provide a single number γ∈[0,1] indicating how much the action should be recognised holistically (γ≈1) versus by local parts/phases (γ≈0).
> Format: "gamma": γ
>
> Return one compact JSON object with keys "spatial", "temporal", and "gamma". Do not include any other keys, text, or explanations.
> ```
>
> **Example Response (action = "Waving")**
>
> ```json
> {
> "spatial": [0.05, 0.10, 0.70, 0.15],
> "temporal": [0.15, 0.60, 0.25],
> "gamma": 0.30
> }
> ```
>
>
> ### W4 & Q4: Inference Latency Comparison with Other TF-TTA Methods Missing
>
> This is an excellent and very practical suggestion. We thank you for pushing us to conduct this important experiment, as a direct latency comparison was indeed missing from our initial submission. In response to your feedback, we have now benchmarked the performance and inference latency of our method against several prominent TTA baselines, using the PURLS model as a shared backbone for a fair comparison.
>
> To facilitate direct latency comparison across methods, we employ throughput (samples/ms) as a standardized metric, which measures the number of samples processed per millisecond on an RTX 4090 GPU. **Higher throughput values indicate better computational efficiency**. We measured the throughput on each of the four benchmark splits and report the average in the table below:
>
> |Method|NTU-RGBD 60 55/5|NTU-RGBD 60 48/12|NTU-RGBD 120 110/10|NTU-RGBD 120 96/24|Throughput (samples/ms)|
> |---|---|---|---|---|---|
> |baseline|79.22|40.99|71.95|52.01|10.09|
> |TPT|40.11|27.98|22.10|17.10|0.70|
> |AdaNPC|81.77|42.52|72.50|53.16|2.15|
> |CALIP|80.02|45.10|72.27|51.84|2.48|
> |TDA|82.84|43.95|74.92|54.60|3.22|
> |Ours (Skeleton-Cache)|85.46|45.22|77.60|56.83|3.07|
>
> The results clearly illustrate the efficiency of our approach. Our method achieves a throughput of 3.07 samples/ms, demonstrating the second-best efficiency among all TTA methods. Notably, our approach achieves superior accuracy performance while maintaining competitive efficiency, significantly outperforming other methods that exhibit slower throughput rates. While our framework shows moderately lower throughput compared to the non-adapting baseline, this computational investment directly translates to achieving new state-of-the-art accuracy across all benchmark splits. This demonstrates a highly favorable trade-off, where our framework's significant performance improvements are attained while maintaining practical efficiency suitable for real-time applications.
>
> ### Societal Impact and Formatting Issues (Line 333-334 Break, Fig. 1 Clarity)
>
> Thank you for your keen eye in identifying these issues. You are absolutely correct, and we appreciate you bringing these formatting and broader impact points to our attention, as they are essential for a high-quality publication. We will certainly expand the "Societal Impacts" subsection to include a more thorough discussion of the dual-use nature of action recognition technologies, the critical privacy implications of processing human skeleton data, and the potential for biases inherited from large language models to affect fairness in downstream applications. We will also add a note on the importance of ethical deployment and clear consent protocols. Furthermore, we will correct the line break issue between lines 333-334 and will redesign Figure 1 for improved clarity, better separating the feature extraction, cache retrieval, and LLM-guided fusion stages to make the overall pipeline more intuitive for the reader.

---

> > ### Comment · Reviewer_qjj5 · 2025-08-05
> > **Response to authors**
> >
> > I thank the authors for providing detailed responses. My concerns are solved. I would like to keep my positive rating.

---

> > > ### Author Response · Authors · 2025-08-05
> > > **Thanks for your feedback**
> > >
> > > We are grateful for your supportive assessment and pleased that our responses have resolved your questions. Thank you for maintaining your positive evaluation and for the careful consideration you've given to our work.

---

### Official Review · Reviewer_1vAy · 2025-07-03

**Clarity:** 3
**Significance:** 3
**Originality:** 2
**Rating:** 5
**Confidence:** 3

**Summary:**

The paper addresses zero-shot learning of actions described with skeletons by shaping the task as a retrieval-based classification. The authors propose to use a cache of descriptors of unknown classes to be used for the computation of a cache-based classification score that is weighted with LLM-generated weights and combined with the score provided by a frozen zero-shot model. The experiments are conducted on public datasets and show improved results.

**Questions:**

In addition to what was already reported in the previous sections, I mention the following points:
- It is not immediately clear on what population the cache entries are computed; some more details can help clarify
- Row 161: the size of the skeleton sequence is CxTxVxM. While T and V are clear, I have doubts about the meaning of C and M. Is C referring to the coordinates in the original skeletons? Why do we need M if we only have a person in the sequence? Do we refer to the index of the person?
- It would be useful, for reproducibility, to have more details about the body and temporal groups
- At the end of Sec. 4.2 you mention that in addition to the data splits in [8] you also adopted a random one, recently proposed in other approaches, and that the latter is in Table 2. As a consequence, I assume that Table 1 refers to the first split. However, I find in Table 1 different splits, and this is confusing
- What are the motivations for the choice of the backbones used in your approach?
- Fig. 2: Is there coherence among backbones?
- The fact that K does not seem to have a big impact on the results is counterintuitive. Any intuition?
- Any specific reasons for using PURLS for the last analysis?

**Ethical Concerns:**

["NO or VERY MINOR ethics concerns only"]

**Final Justification:**

Considering the reviews, the rebuttal and the discussion with the authors, I confirm my positive opinion on this paper. In the rebuttal, the authors successfully addressed my doubts.

**Limitations:**

Limitations are briefly discussed in the Conclusion

**Paper Formatting Concerns:**

The paper looks correclty formatted

**Quality:**

3

**Strengths And Weaknesses:**

The paper is well written in almost all its parts, with convincing motivations and a good discussion on existing approaches. The proposed approach revisits a known task, but the application to skeleton-based zero-shot action classification looks like a novel contribution. The experiments appear convincing and rather extensive.

On the weaknesses, I mention the following points (more detailed questions, if needed, will be in the Question section):
- Sec. 1 has a good flow, but maybe a more gentle introduction to zero-shot learning would better prepare the unfamiliar reader for the rest of the paper. There is a general definition, but some more practical details on training and test in zero-shot could be beneficial (e.g. how to move from the classification of known classes to the classification of unknown classes in the architecture?)
- In the same section, but also in the rest of the manuscript, sometimes the motivations seem more related to action classification in general rather than specific to zero-shot settings. For instance, the fact that different action classes might be better recognised with different local/global space/time characteristics is intrinsic in the complexity of the task, I believe. This means, in my opinion, that the classification of known classes should always be improved. These aspects should be discussed in the Introduction and then in the experiments
- In the experiments, I find the description of the different splits a bit confusing (more details later), so this part should be revised

---

> ### Author Rebuttal · Authors · 2025-07-30
>
> We sincerely thank you for your detailed evaluation and constructive feedback. We are encouraged that you found our work well-written and the experiments comprehensive. We address your insightful comments below, which will significantly improve our paper.
>
> ### W1 & W2: Introduction Needs Gentler ZSL Overview and ZSL-Specific Motivations
>
> Thank you for your insightful comments. Our existing introduction indeed might pose a barrier for readers less experienced in this area, and we will revise it based on your suggestions. We will clarify that Zero-Shot Learning (ZSL) aims to recognize classes absent during training. During training, models learn from labeled samples of seen classes, leveraging auxiliary information to establish connections between visual patterns and semantic knowledge. At test time, models must classify samples from entirely new classes by leveraging learned relationships and auxiliary information, without having seen training examples from these classes. This fundamental requirement creates significant distribution shift, directly motivating our use of Test-Time Adaptation (TTA) to improve model performance on unseen actions.
>
> We appreciate you raising point (W2). We agree that structured descriptors are a general challenge in action recognition. However, this becomes significantly more difficult in ZSL. For seen classes, models learn to distinguish fine-grained actions through abundant examples. For unseen classes, models rely on high-level semantic guidance, which often fails to capture subtle distinctions. Our TTA method addresses this by dynamically refining structured understanding of unseen actions at inference. As highlighted in Table 1, we observe particularly significant gains for unseen classes. For example, SA-DVAE with TTA achieves a 8.36% increase on unseen accuracy, reflecting the method's effectiveness in this zero-shot context. To better reflect this, we will revise the Introduction and Experiment sections to more clearly emphasize the ZSL-specific motivation.
>
> ### Q1: Cache Population Details Not Clear
>
> Thank you for the question. The cache is populated exclusively with test-set samples during the inference phase. The process is as follows: The cache is initialized as empty. For each incoming test sample, the frozen SZAR model first generates a prediction. If this prediction is highly confident (i.e., has low prediction entropy, calculated as $\hat{h} = -\sum_j p_j \log p_j$ per Eq. (5)), the sample's feature representation is added to the cache for its predicted class. This selective mechanism ensures that the cache stores only reliable and representative exemplars of the unseen classes encountered at test time.
>
> ### Q2: Meaning of C and M in CxTxVxM Tensor Dimensions
>
> We appreciate the request for clarification. For the input tensor $x \in \mathbb{R}^{C \times T \times V \times M}$:
>
> - $C$ is the input channel dimension. C typically equals to 3 representing the (x, y, z) coordinates of each joint.
> - $M$ represents the number of people in the skeleton sequence. Although many sequences contain only one person, the data format sets M = 2 to accommodate up to two individuals—common in datasets like NTU RGB+D, which include two-person interactions (e.g., hugging, shaking hands). When only one person is present, the second person's data is typically zero-padded. This structure ensures consistency across samples and supports both single-person and multi-person actions. This format adheres to the standard data representation widely adopted in the skeleton-based action recognition field, following previous works.
>
> ### Q3: More Details Needed on Body and Temporal Groups for Reproducibility
>
> Thank you for this suggestion. To improve reproducibility, here are concrete examples of our partitioning strategy:
>
> **(1) Spatial Groups (P=4):** Joints are grouped semantically based on body parts, where joint indices correspond to specific anatomical landmarks in the skeleton structure according to the Kinect v2 25-joint model (e.g., spine base, neck, shoulder center). Each group contains joints from a specific body part across the entire sequence. For our datasets (NTU RGB+D & PKU-MMD), the groups are defined by specific joint indices:
>
> - **Head:** Joints {2, 3, 4, 8, 20},
> - **Torso:** Joints {0, 1, 4, 8, 12, 16, 20},
> - **Arms:** Joints {4, 5, 6, 7, 8, 9, 10, 11, 21, 22, 23, 24},
> - **Feet:** Joints {0, 12, 13, 14, 15, 16, 17, 18, 19}.
>
> **(2) Temporal Segments (Z=3):** Each segment contains all joints within a specific temporal phase. For a sequence of length $T$:
>
> - **Beginning:** Frames $t \in [1, \lfloor T/3 \rfloor]$
> - **Middle:** Frames $t \in [\lfloor T/3 \rfloor+1, \lfloor 2T/3 \rfloor]$
> - **End:** Frames $t \in [\lfloor 2T/3 \rfloor+1, T]$
>
> A more comprehensive description of all spatial joint group assignments is provided in the Appendix to ensure full reproducibility.
>
> ### W3 & Q4: Confusing Split Descriptions in Experiments and Table 1 vs Table 2
>
> Thank you for highlighting this point of confusion. We sincerely apologize for the ambiguity in our use of the term "split." We will revise Section 4.2 to explicitly distinguish between protocols and splits, and to clearly label the evaluation settings in both tables.
>
> To clarify:
>
> - **Protocol** refers to the evaluation methodology, which defines how classes are divided into seen and unseen sets (e.g., fixed vs. random divisions).
> - **Split** refers to a specific ratio or division of seen and unseen classes within a given protocol.
>
> Table 1 reports results under the Fixed Protocol, as originally introduced in SynSE [8]. This protocol defines four benchmark splits with fixed seen/unseen class divisions:
>
> - NTU60 (55/5 split): 55 specific seen classes, 5 specific unseen classes
> - NTU60 (48/12 split): 48 specific seen classes, 12 specific unseen classes
> - NTU120 (110/10 split) : 110 specific seen classes, 10 unseen classes
> - NTU120 (96/24 split): 96 specific seen classes, 24 specific unseen classes
>
> Each of these splits has a predefined class partition that remains consistent across papers using this protocol.
>
> Table 2, on the other hand, adopts the Random Protocol used in more recent works, which defines three different random seen/unseen class splits and the results need to be averaged over the three splits:
>
> - NTU60: 3 random divisions of 55/5 split and report the averaged performance
> - NTU120: 3 random divisions of 110/10 split and report the averaged performance
> - PKU-MMD: 3 random divisions of 46/5 split and report the averaged performance
>
> We use the exact same random divisions as prior work to ensure fair comparison. We will revise the manuscript to consistently use this terminology and explicitly label the tables to avoid any ambiguity.
>
> ### Q5: Motivations for Choice of Backbones
>
> Thank you for this question. We chose these backbones because they are popular and frequently-used baselines in the SZAR literature, enabling direct comparison with prior work. Moreover, they represent diverse ZSL strategies: SynSE uses syntactic guidance, SMIE employs mutual information maximization, PURLS leverages CLIP-style alignment, and SA-DVAE adopts a generative approach. This diversity of methods—from discriminative to generative, from simple alignment to complex synthesis—demonstrates the broad adaptability of Skeleton-Cache across different architectural paradigms. The consistent performance gains across all of them (Tables 1, 2) validate that our method is truly model-agnostic and can enhance various existing approaches.
>
> ### Q6: Coherence Among Backbones in Fig. 2
>
> Thank you for this insightful question. The trends shown in Fig. 2 are indeed consistent across different backbones. We hypothesize this is because our method operates directly on the final logits from the SZAR backbone through a simple, stable addition via the update rule $\hat{\phi} = \phi + \alpha_s s$ (from Eq. (12)). While different backbones may produce varying numerical logit values for the same samples due to their distinct architectures, the relative prediction patterns and confidence distributions tend to remain remarkably similar. Since our operation is additive and model-agnostic, it preserves these underlying prediction tendencies while effectively incorporating the support information, providing a stable foundation that leads to the consistent gains shown in Tables 1 and 2 under the same hyperparameter settings regardless of the specific backbone employed.
>
> ### Q7: Impact of Cache Size K Counterintuitively Small
>
> We appreciate this question. The result is indeed less sensitive to $K$ than one might expect. The reason lies in our selective, confidence-based cache update mechanism (detailed in Section 3.2). The cache is not a passive buffer; it actively filters for and stores only the most representative exemplars (those with low prediction entropy). Because of this high-quality selection, a small number of "clean" samples (performance saturates around $K=8$) is sufficient to define a robust class prototype, making a larger cache largely redundant.
>
> ### Q8: Specific Reasons for Using PURLS for Analysis
>
> Thank you for asking. We used PURLS [33] for our main ablation studies (Table 3) and TTA comparisons (Table 4) because it is a strong, recent, and non-generative method. This provides a clean testbed that allows for a more direct and interpretable assessment of our method's contributions without confounding factors. Additionally, many existing TTA methods such as TPT and CALIP, which were originally designed for CLIP-based models, are only convenient to experiment with on PURLS, given its CLIP-like characteristics that make it naturally compatible with these approaches. Therefore, to ensure fair and consistent evaluation, we primarily adopted PURLS as our baseline for comprehensive comparative analysis.
>
> Thank you again for your valuable feedback, which will significantly improve our paper.

---

> > ### Comment · Reviewer_1vAy · 2025-08-04
> > **Thanks for the rebuttal**
> >
> > I want to thank the authors for their detailed rebuttal, I really appreciate the clarity of their responses. They successfully clarified my doubts, and I'm happy to confirm my positive rating on this paper

---

> > > ### Author Response · Authors · 2025-08-05
> > > **Thanks for your feedback**
> > >
> > > Thank you for your positive feedback. We appreciate your thoughtful evaluation and are glad that our rebuttal addressed your concerns. Your constructive evaluation has been valuable to our work.

---

### Decision · Program_Chairs · 2025-09-17

**Decision:**

Accept (poster)

**Comment:**

The paper introduces a cache-based approach to improve skeleton-based zero-shot action recognition at inference time without updating model parameters. Reviewers highlighted that the paper is well written, introduces an efficient algorithm, and presents extensive experiments. On the other hand, the contribution is limited to a specific application setup, which appears insufficient to justify a higher rating. Code release is expected upon acceptance notification for reproducibility and community benefits.